# Potentiators empower synthetic microbiomes as silent guardians against co-contamination

Zhepu Ruan ®[1,4], Jialin Tan ®[1,4], Qingyu Feng ®[1], Kaiqing Yang ®[1], Danning Li[1], Yuanqing Chao ®[2], Peng Wang ®[2], Zhuobiao Ni ®[1], Jingjing Chen[1,3] & Rongliang Qiu ®[1,2] ✉

Environmental co-contamination presents significant challenges. To tackle these, while microbial consortia offer advantages over single-strain, such as functional redundancy and synergistic degradation, rationally designing effective synthetic microbiomes for complex co-contamination scenarios remains a major challenge. Here, we utilize our advanced genome-scale metabolic modeling (GSMM) tool, SuperCC, to simulate the metabolic behavior of communities consisting of six isolated key strains under single- and multi-carbon source conditions, mimicking single-pollutant or co-contamination scenarios respectively. By integrating multi-omics data with metabolic modeling of cultures, we systematically elucidate key strain interaction networks and adaptive strategies under co-contamination. This reveal that the specific secretory products of broad-spectrum resource-utilizing bacteria serve as key metabolites driving cooperation and highlight the pivotal role of indigenous keystone strains in stabilizing and enhancing community function. Consequently, we propose an innovative and rational paradigm for consortium design: DHP-Com (Degrader-Helper-Potentiator Consortium). Potentiators are top species with stable habitat abundance. Synthetic microbiomes built on this framework exhibit enhanced ecological fitness and substantially improve remediation performance across diverse co-contamination scenarios. Our findings advanced the practical application of GSMM predictions to decipher intricate multi-pollutant/multi-strain interaction networks, offering a powerful rational framework and robust methodological tools for engineering multi-functional and effective synthetic microbiomes for complex environmental remediation.

The escalating coexistence of multiple organic pollutants[1,2]—driven by modern industrial and agricultural activities—poses a critical challenge for global environmental remediation[1]. Compared with single-pollutant scenarios, co-contamination presents greater difficulties due to complex compositions, poorly understood interactions, and unpredictable environmental behaviors[2]. Traditional physicochemical methods are costly and may cause secondary pollution, whereas microbial remediation offers a more sustainable and eco-friendlier

[1]Guangdong Laboratory for Lingnan Modern Agriculture, Guangdong Provincial Key Laboratory of Agricultural & Rural Pollution Abatement and Environmental Safety, College of Natural Resources and Environment, South China Agricultural University, Guangzhou, China. [2]School of Environmental Science and Engineering, Sun Yat-sen University, Guangzhou, China. [3]College of Urban and Environmental Sciences, Hubei Normal University, Huangshi, China. [4]These authors contributed equally: Zhepu Ruan, Jialin Tan. ✉e-mail: qiurl@scau.edu.cn

alternative[3,4]. Although many pure strains capable of degrading specific pollutants have been identified, their limited adaptability hampers their effectiveness in complex environments[5]. Moreover, simply mixing degraders can lead to metabolic competition and ecological conflicts[6,7], underscoring the urgent need for coordinated and efficient microbial strategies.

Microbiomes provide a promising alternative to overcome the limitations of single-strain bioremediation. Under pollutant stress, microbiomes perform metabolic division of labor, signal transduction, and resource exchange, enhancing degradation and adaptability[8–10]. For instance, cross-feeding between strains can significantly boost degradation efficiency, supporting the Degrader&Helper model[11,12]. While studies have shown that narrow-spectrum resource-utilizing bacteria drive the stability of synthetic communities for plant growth promotion[13], research on keystone strains that stabilize community structure in co-contaminated environments remains scarce−particularly regarding the undefined roles of indigenous microbes without direct/indirect degradative functions in responding to diverse pollutants, thereby limiting the development of functionally stable and effective consortia[14]. Moreover, the limited survival of individual degraders in complex environments, coupled with the poorly understood functional roles within indigenous microbiomes, has hindered the development of environmentally resilient, functionally stable consortia based on native microbial communities.

Synthetic microbiome design currently follows two strategies: top-down enrichment of the natural microbiome into a functional microbiome, and bottom-up assembly of keystone strains into simplified communities[15,16]. However, both approaches depend on traditional microbial screening methods, resulting in inefficient consortium construction driven by empirical trial-and-error approaches. Recent advances in genome-scale metabolic models (GSMMs) have enabled systematic exploration of metabolic networks through gene-enzyme-reaction mapping[17,18]. Nevertheless, existing studies primarily focus on single-pollutant scenarios, whereas microbiome design for co-contamination requires explicit consideration of cross-species metabolic interactions and niche partitioning[11]. Additionally, it is important to note that a key limitation in applying GSMMs to multi-strain communities lies in the difficulty of systematically exploring all possible strain combinations and balancing their cooperative and competitive interactions. To address this, constraint-based approaches that can navigate the vast design space of complex communities without predefining interaction types are needed. Currently, no systematic framework exists for the GSMM-guided design of synthetic microbiomes under multi-pollutant stress, particularly one that integrates native microbial functionalities with engineered community-level stability. To bridge this gap, we introduce our Super Community Combinations (SuperCC) framework as a prelude, which is designed to address this exact need by enabling such comprehensive analysis without limitations on microbiome size[7].

In this study, we demonstrate how the SuperCC framework enables the systematic design of synthetic microbiomes for co-contamination. Specifically, we establish single- and multi-carbon culture systems to mimic single and compound pollution scenarios, thereby extending SuperCC's applicability and demonstrating its potential to link theoretical predictions to ecological complexity. By combining multi-omics and metabolic modeling, we uncover key strain interactions and adaptation strategies under compound pollution (Fig. 1a). Our work identifies the pivotal role of indigenous stabilizing strains, establishing an innovative DHP-Com (Degrader−Helper−Potentiator Consortium) paradigm for microbiome construction, and also introducing a Potentiator Contribution Index (PCI) to quantitatively measure the potentiator's effect across conditions (Table 1). This approach enhances survival and pollutant removal in complex contamination. These findings advance GSMM-based multi-pollutant/multi-strain network predictions, offering theoretical and practical tools for synthetic microbiome engineering.

## Results
### Co-contamination enhances biodegradation and microbial survival
Water samples were collected from a wastewater treatment plant, and after ten rounds of gradient acclimation, three functional microbial consortia were obtained: tetracycline (TC), oxytetracycline (OTC), and a combination of both (TCs) ("Methods", Fig. 1a). To quantify degradation efficiency, we used HPLC-MS and found that the TCs consortium from the tenth acclimation generation showed superior degradation efficiency of tetracycline antibiotics, particularly OTC, compared to the single-pollutant groups (one-way ANOVA, $p = 0.00098$; Fig. 1b). We then used 16S rRNA amplicon sequencing to explore whether this enhanced degradation was linked to community structure changes. Relative to the TC group, the TCs consortium exhibited significantly higher $\alpha$-diversity (one-way ANOVA, $p = 0.03411$), suggesting co-contamination alleviates the negative effects of TC on microbial communities (Fig. 1c). $\beta$-diversity analysis further confirmed that pollutant type drove community differentiation: despite sharing the same origin, the three consortia formed distinct clusters under different stressors (Fig. 1d). Functional predictions from 16S rRNA profiles provided mechanistic insights: while amino acid and carbohydrate metabolism pathways dominated all groups (Supplementary Fig. 1), the TCs consortium showed specific enrichment in xenobiotic biodegradation pathways (e.g., xylene, dioxin, and polycyclic aromatic hydrocarbon degradation)−even as its abundance in most other metabolic pathways decreased relative to single-pollutant groups. Notably, key genes involved in aromatic compound degradation (critical for tetracycline breakdown) were significantly upregulated in the TCs (Fig. 1e and Supplementary Fig. 2a, b), directly linking community adaptation to enhanced degradation function.

### Modeling the Top-50 species to narrow down the functional microbiome
The observed changes in community structure and function raised a follow-up question: which species within the consortium drive these traits? We focused on the Top-50 most abundant bacteria in each consortium, as these are likely key contributors to ecosystem function. Since the consortia shared the same origin, the Top-50 strains overlapped substantially, resulting in 70 unique strains after deduplication (Supplementary Data 1). We constructed single-strain metabolic models to characterize their ecological strategies under complete medium (CM). Correlation analysis revealed three key patterns: biomass production correlated positively with growth time (Student's $t$-test, $p = 0.0107$, Fig. 2a), total metabolic reactions (Student's $t$-test, $p < 0.0001$, Fig. 2b), and exchange reactions (Student's $t$-test, $p < 0.0001$, Fig. 2c). This indicated that strains with more extensive metabolic networks produced greater biomass but slower growth. The exchange reaction also reflects the resource utilization spectrum of the strain[13]; thus, the broader the range of resources that microorganisms can utilize, the greater the biomass of their single cells.

When we compared traits under different stressors, we observed clear selective pressures: TC stress favored strains with biomass ranging from 130 to 210 mmol/g DW (Fig. 2d), OTC selected for high-biomass, slow-growing strains (up to 356.5305 mmol/g DW and 0.054 s, Fig. 2e), and co-contamination (TCs) specifically enriched fast-growing, low-biomass strains (Fig. 2f and Supplementary Fig. 3). Hierarchical stratification further refined these patterns: under OTC, the top 10 most abundant species had low biomass and, narrow resource use, and fast growth (Fig. 2g), while mid-ranking strains (positions 11–30) had enhanced metabolic interaction potential (more exchange reactions). In contrast, the TC and TCs groups showed opposing patterns. Across all groups, lower-ranked strains (positions 31–50) consistently had broader resource use and higher exchange capacity than higher-ranked counterparts (Fig. 2h). Correlation

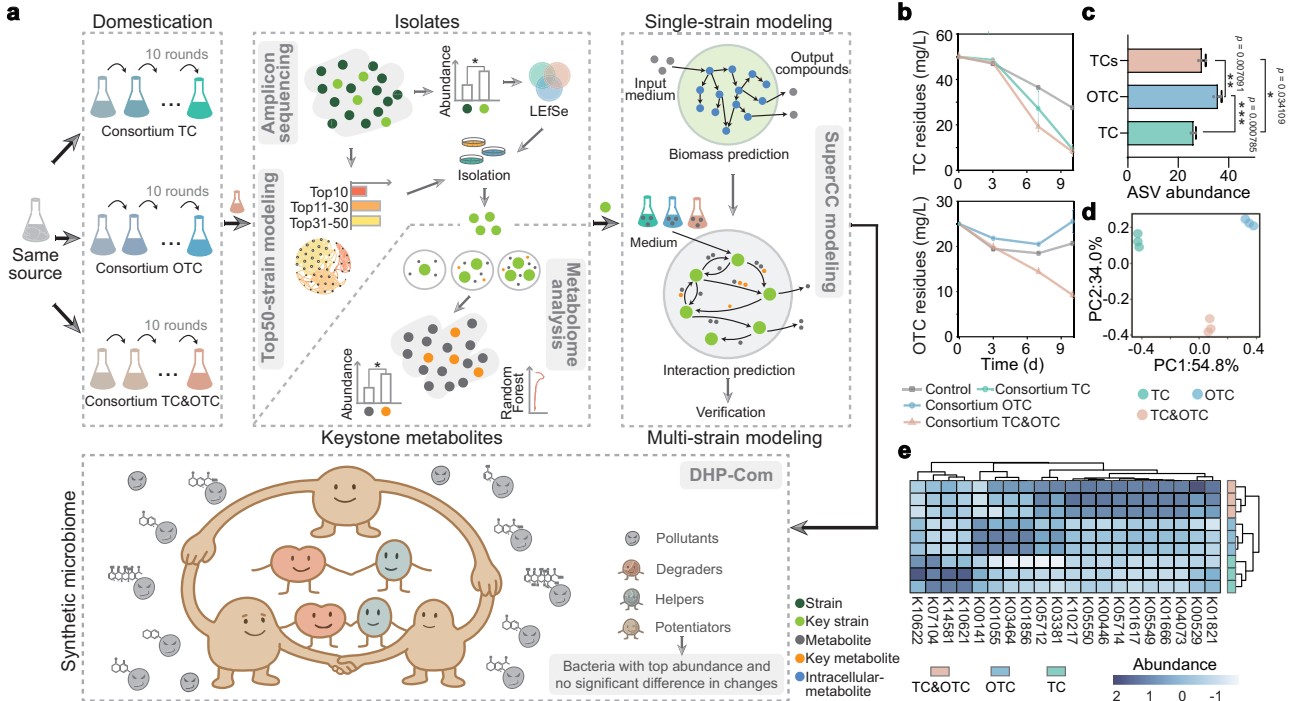

**Fig. 1 | Experimental design and microbial consortium responses to combined antibiotic pollution. a** Workflow for domesticating aquatic microbial communities from the same water sample via exposure to tetracycline (TC), oxytetracycline (OTC), and their combination (TCs). Functional microbiomes were analyzed by amplicon sequencing and metabolic modeling of the top 50 abundant strains. Potential keystone species were thus identified by integrating abundance shifts and strain isolation. Untargeted metabolomics was applied to single-, co-, and tri-strain cocultures under three treatments: antibiotic-free control (CK), exposure to a single antibiotic (TC or OTC), and exposure to the combined TCs treatment. Since metabolomic analysis was only performed on co-cultures with up to three strains, the microbial community modeling framework SuperCC was applied to construct multi-strain metabolic models with a larger number of strains, aiming to predict strain-specific biomass and identify key interspecies metabolites. Model predictions were subsequently validated through experimental analyses. Based on these conclusions, we proposed a synthetic microbiome construction strategy: the Degrader–Helper–Potentiator Consortium (DHP-Com) paradigm, and provided the

definition of potentiator. **b** Enhanced degradation efficiency of TC and OTC by acclimated microbial consortia. The degradation efficiency was determined by measuring residual antibiotic concentrations in degradation medium (DM) supplemented with 50 mg/L TC or 25 mg/L OTC at 0, 3, 7, and 10 days after inoculation with the consortia acclimated over 10 rounds. Data are presented as mean values ± SD ($n = 3$ biological independent replicates). **c** The Chao1 index was used to evaluate microbiome $\alpha$-diversity within each consortium. Statistical significance of observed differences was determined by one-way ANOVA (*$p < 0.05$, **$p < 0.01$, ***$p < 0.001$, n = 3 biological independent replicates). Data are presented as mean values ± SD. **d** $\beta$-diversity was assessed through Principal Coordinates Analysis (PCoA) using Bray-Curtis dissimilarity metrics. **e** Gene profiles related to aromatic compound degradation. The relative abundance of key aromatic degradation genes at the KEGG Orthology (KO) level across treatments was visualized by heatmap, with functional predictions generated using PICRUSt2. Source data for this figure is available in the Source data file.

network analysis added community-level context: TC communities showed the highest edge density, OTC exhibited the lowest modularity but highest average node degree, and TCs exhibited the highest modularity but lowest edge density (Fig. 2i). The high modularity of TCs suggested co-contamination favors functionally specialized strains, with modularity supporting efficient metabolic coordination at the community level.

**Identifying keystone species to streamline degrading consortia**
To translate these community-level patterns into practical applications (e.g., streamlined degrading consortia), we next isolated key strains from the three consortia. We obtained six strains: *Alcaligenes* sp. A1, *Providencia* sp. P1, *Comamonas* sp. C2, *Leucobacter* sp. L3, *Brevundimonas* sp. B4, and *Stenotrophomonas* sp. S4 (Fig. 3a, c). Their distribution reflected stress-specific selection: A1 and P1 were unique to TCs, B4 and S4 were present in all consortia, and L3 was exclusive to OTC (and thus excluded from subsequent combinations due to its narrow habitat range).

We tested their degradation and growth synergies in single-, co-, and tri-culture experiments. After 4 h of cultivation, single strains C2, L3, and S4 exhibited positive growth. However, growth dynamics changed in co-cultures: strains A1 or P1 enhanced growth in pairwise combinations, while tri-cultures with B4 and S4 demonstrated

synergistic growth (Fig. 3b). For degradation, only C2 and S4 degraded both TC and OTC, while P1 degraded TC via co-metabolism (Supplementary Fig. 4). In mineral salts medium (MM, no additional carbon sources), only the A1/P1/C2 co-culture showed enhanced degradation under co-contamination relative to single-pollutant conditions. Notably, tri-cultures consistently outperformed single/dual cultures in pollutant removal: the A1&C2&B4 consortium achieved the highest OTC (8.87%) and TC (5.05%) removal rates under co-contamination (Fig. 3d).

We further validated these strains' ecological relevance by quantifying their abundance in the original consortia—their distribution matched their isolation sources (Fig. 3e). Genome-scale metabolic models (manually curated) revealed functional differences: strains A1 and P1 had 1.42-fold and 1.34-fold more metabolic reactions (especially exchange/transport reactions) than the other strains, confirming their broad resource utilization. Growth simulations under varying nutrient conditions showed that B4 and S4 achieved comparatively greater biomass under oligotrophic conditions (with only one carbon source) (Fig. 3f). Venn diagram analysis of exchange metabolites identified 35 compounds shared among all six strains, with A1 and P1 contributing unique metabolites (amino acids (AAs), sulfonates, carbohydrates, nucleotides, vitamins, Fig. 3g; see Supplementary Table 1 for details).

**Table 1 | Characteristics of degrader, helper, potentiator, and Potentiator Contribution Index (PCI) in DHP-Com**

| | Abundance | Significant difference in changes | Function | In this study |
|---|---|---|---|---|
| Degrader | Not necessarily high | Yes | Direct function, degrades pollutants | Strain C2 |
| Helper | Not necessarily high | Yes | Indirect function helps degraders | Strains A1, P1, L3 |
| Potentiator | High | No | Indirect function helps degraders and helpers | Strains B4, S4 (The abundance is usually among the top 30). |
| Potentiator Contribution Index (PCI) | $PCI = \dfrac{Perfprmance(DHP) - Perfprmance(DH)}{Perfprmance(DH)} \times 100\%$ | | | |

Perfprmance(DHP) = Overall performance (e.g., pollutant degradation rate, community biomass) of the consortium containing *Degrader (D), Helper (H),* and *Potentiator (P).*
*Perfprmance (DH)* Performance of the consortium with only *Degrader (D)* and *Helper (H).*

### Identifying key metabolites to explore microbial interactions

The synergistic growth and degradation of keystone species suggested they rely on metabolite-mediated interactions. To test this, we performed untargeted metabolomic analysis on single-, co-, and tri-strain cultures exposed to three stress conditions: antibiotic-free (CK), single-exposure (TC/OTC), and combined-stress (TCs) (Methods). Principal component analysis (PCA) revealed a critical pattern: stressor type (not strain combination count) determined metabolite clustering. Notably, communities containing B4/S4 exhibited similar metabolite profiles, whereas the broad-resource-utilizing A1/P1/C2 consortium formed a distinct cluster—especially under co-contamination (Fig. 4a). Quantitative analysis showed that relative to CK, all groups had more types of secreted metabolites and higher abundances under TCs (Student's t-test, two-tailed, $p < 0.05$, VIP value $\geq 1.00$). Under TCs, multi-strain consortia showed greater secretory diversity (facilitating stress sharing), whereas single/dual-strain cultures secreted fewer metabolite types but higher upregulation fold-changes. Under OTC alone, dual-strain combinations exhibited comparatively stronger metabolic interactions. Among all tested groups, the A1&P1&C2 tri-strain consortium and its pairwise combinations consistently demonstrated the highest diversity of secreted metabolites and the greatest fold-changes in upregulation (Fig. 4b).

To identify key secreted metabolites, we applied a more stringent significance threshold (Student's t-test, $p < 0.01$, VIP value $\geq 2.00$). Across both single- and multi-stressor conditions, the tri-strain consortium exhibited the highest number of unique metabolites (Fig. 4c and Supplementary Fig. 5). Under combined pollution, 308 metabolites were commonly secreted by all three microbial combinations. Additionally, the tri-strain and single-strain cultures produced a greater number of unique metabolites (103 and 56, respectively). Comparison with the Human Metabolome Database (HMDB) revealed that many detected metabolites were unannotated or not previously reported, indicating their potential as novel or stress-specific compounds. Among the annotated metabolites, those significantly upregulated—whether unique to the tri-strain group or shared—were primarily classified as AAs, peptides, and analogues, followed by carbohydrates, constituting core metabolic features (Fig. 4c and Supplementary Fig. 6). Random forest analysis of three unique secreted metabolites identified highly influential compounds likely involved in stress responses (e.g., oxidative stress, detoxification) or key regulatory pathways (Fig. 4c). For statistics of differential metabolite screening under multiple statistical criteria (including FDR correction), see Supplementary Data 2.

### Modeling inter-species interactions among keystone species

To deepen our understanding of metabolite-mediated interactions, we used the SuperCC modeling tool to simulate synthetic microbiomes (1–6 keystone strains). Since no single strain could utilize TCs as the sole carbon source, microbial cooperation depended on degrading pollutants into utilizable metabolites, with co-contamination providing more diverse carbon sources. We simulated single- ($C = 1$) and co-pollution ($C = 4$) using different carbon substrates. Flux balance analysis (FBA) revealed a binary classification based on B4 presence (Fig. 5a): B4-containing consortia showed significantly higher biomass (despite B4's poor monoculture performance), whereas B4-absent groups displayed reduced biomass with increasing carbon availability. We further simulated extreme biomass optimization scenarios where metabolic resources in the community were exclusively allocated to each target strain. The results showed that maximum strain-specific biomass was genetically constrained (invariant to $C = 1/C = 4$ conditions), though growth rates varied significantly (Fig. 5b). Most strains grew faster when exposed to multiple carbon sources, particularly P1 ($0.005375\,s^{-1}$ faster in $C = 4$), while B4 exhibited the opposite trend ($0.006375\,s^{-1}$ slower), explaining their differential survival under single or two pollutants.

FBA of interspecies metabolic interactions revealed AAs and organic acids (OAs) as the primary exchange metabolites, with A1, P1, and C2—the Degrader&Helper combination—forming a central interactive triad (Fig. 5c). These core strains demonstrated maximal metabolic dependence on others under $C = 1$ conditions, although such cooperation is likely limited by resource competition in natural ecosystems. The reduction in metabolite exchange observed under $C = 4$ conditions closely corresponded to the selective enrichment of A1 and P1 specifically under TCs stress. Additionally, strain L3 was less abundant and exhibited minimal metabolic interaction with relatively few partners (Supplementary Fig. 7). Most notably, B4 consistently demonstrated non-competitive cross-feeding behavior across all tested conditions, suggesting its role as an altruistic keystone species essential for maintaining community stability (Fig. 5c and Supplementary Data 3). All condition-specific differences in exchanged metabolites (e.g., metabolites unique to $C = 1$ vs. $C = 4$) are annotated in the Source Data. For flux balance analysis illustrating interspecies metabolic exchange under the objective of equalizing biomass across species, see Supplementary Data 4.

### Testing potentiator's metabolic advantages under co-contamination

Finally, we tested whether keystone species and their metabolites could enhance resilience under real-world, multi-pollutant stress. We exposed the six-strain consortium to eight pollutants: biocides (dodecyl trimethyl ammonium chloride, DTAC; dodecyldimethylbenzylammonium chloride, DBAC), herbicides (bromoxynil octanoate, BO; quinclorac, QC), sulfonamide antibiotics (sulfamethoxazole, SMX; sulfamethazine, SMT), and tetracyclines (TC, OTC). To elucidate the functional contributions of strains B4 and S4, we compared the full six-strain consortium with two sub-consortia: one comprising B4 and S4, and another comprising A1, P1, C2, and L3. After 2 days of cultivation, the full consortium outperformed both sub-consortia under co-contaminant conditions (Fig. 6b), confirming the non-redundant roles of all keystone strains. To clarify the criteria for including exchange metabolites in our analyses,

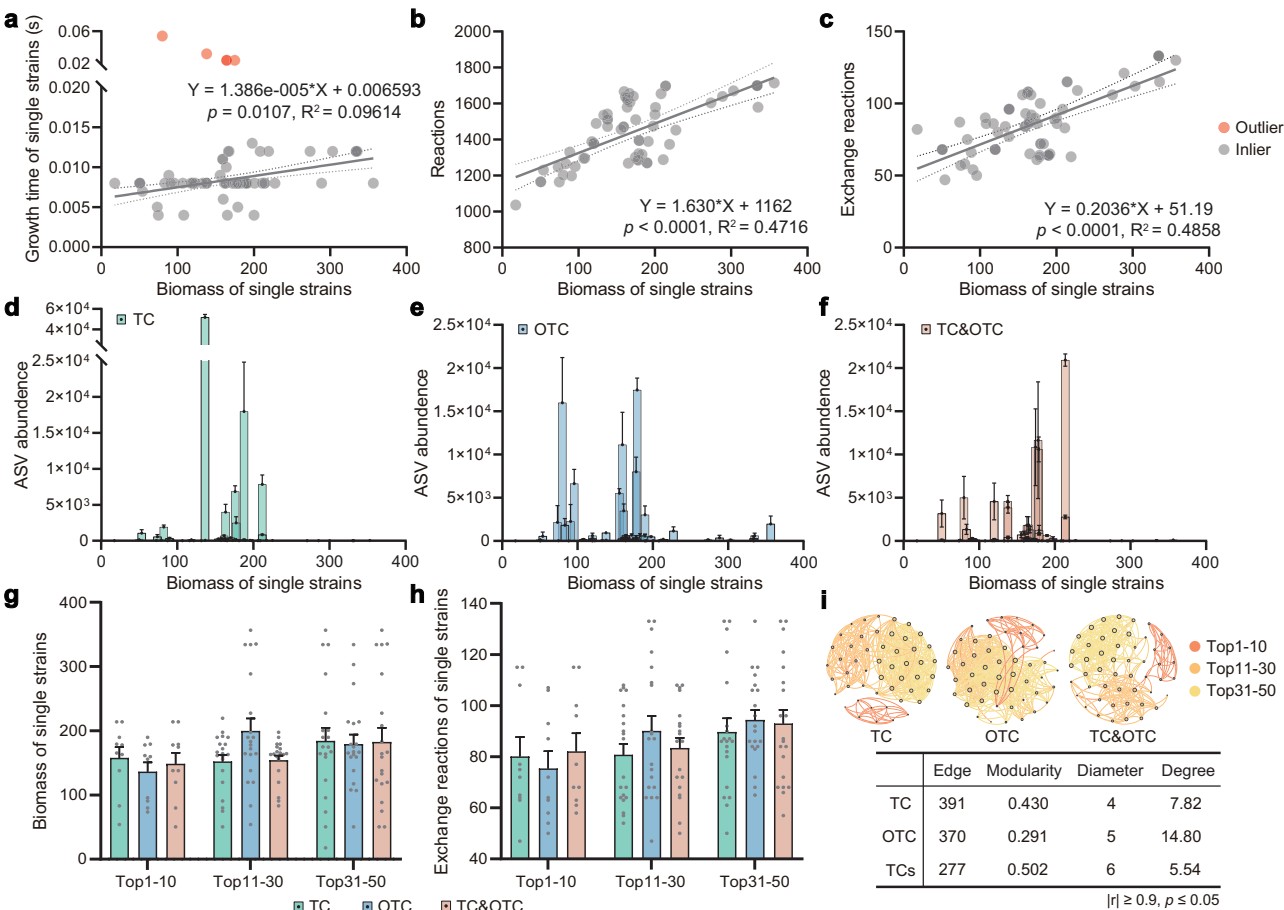

**Fig. 2 | Computational modeling of dominant microbial species within acclimated consortia. a** Simulated relationship between growth time and biomass production in complete medium. Outliers (red dots) with excessive variability in growth time were excluded from the trend analysis (Student's *t*-test, simple linear regression). **b** Relationship between total metabolic reactions and biomass production in complete medium (Student's *t*-test, simple linear regression). **c** Relationship between the number of exchange reactions and biomass production in complete medium (Student's *t*-test, simple linear regression). Comparative analysis of model-predicted biomass and experimentally observed relative abundance for the Top-50 species in the TC consortium (**d**), OTC consortium (**e**), and TC&OTC consortium (**f**) (*n* = 3 biological independent replicates). Data are presented as mean values ± SEM. **g** Biomass production potential stratified by species rank: Top1-10, 11-30, and 31-50. Data are presented as mean values ± SEM. **h** Number of exchange reactions across ranking tiers. Data are presented as mean values ± SEM. **i** Topological analysis of species co-occurrence networks among the Top-50 species across all three acclimation conditions. Nodes represent species (ASVs), with node size proportional to its degree (number of connections). Edges represent statistically significant strong correlations ($|r| \geq 0.9$, $p \leq 0.05$) between species, with their color corresponding to the abundance rank category of the connecting nodes. The table below the network panels provides a summary of the key topological parameters for each of the three scenarios. TCs, TC&OTC. Data are presented as mean values ± SD. Source data for this figure is available in the Source data file.

we detail the evidence from metabolomics, single-strain, and multi-strain modeling in Supplementary Table 2. Cofactor supplementation was tested under these stresses. After monitoring every 15 min at 30 °C for 1800 min using a microplate reader, we found that, despite mediating intensive cross-feeding, amino acids showed minimal adaptation benefits, with biomass matching controls or exhibiting inhibition (Fig. 6a). Other metabolites consistently enhanced survival. Keystone strains (TCs-enriched) demonstrated significantly enhanced growth when supplemented with predicted metabolites (one-way ANOVA, $p < 0.01$). The complete consortium exhibited notable resilience, achieving substantial pollutant removal under high-intensity co-contamination, even though individual strains lacked the capacity to degrade biocides, sulfonamides, or herbicides. This performance was particularly evident with metabolite supplementation (Supplementary Fig. 8a); BO degradation was evaluated via the residual levels of its intermediate metabolite, bromoxynil. Vitamins alone maximized survival under sulfonamide/herbicide stress (with higher Area Under Curve values), while biocides showed improved late-phase recovery only with full metabolite complementation. These findings underscore the

essential and non-redundant roles of keystone strains in sustaining microbial resilience through metabolic cooperation, even under the burden of eight simultaneous pollutants (Supplementary Fig. 8b). To further clarify the quantitative definition of a potentiator, we introduce a Potentiator Contribution Index (PCI) and present PCI values for microbial growth and degradation rate in Table 1 and Supplementary Table 3.

## Discussion

In this study, we employed a gradient domestication approach, exposing microbial communities to both antibiotics while monitoring residual concentrations. Unexpectedly, co-contamination-adapted communities exhibited enhanced degradation and greater degrader diversity, which aligns with Smith et al.[2], who observed improved microbial growth under combined stress. This prompted us to investigate whether co-contamination offers degradation advantages. Traditional amplicon sequencing-based correlation networks are limited to abundance and taxonomy, failing to reveal functional potential. To decode microbial interactions, we employed a multi-omics-integrated GSMM approach (using the SuperCC tool) to analyze six

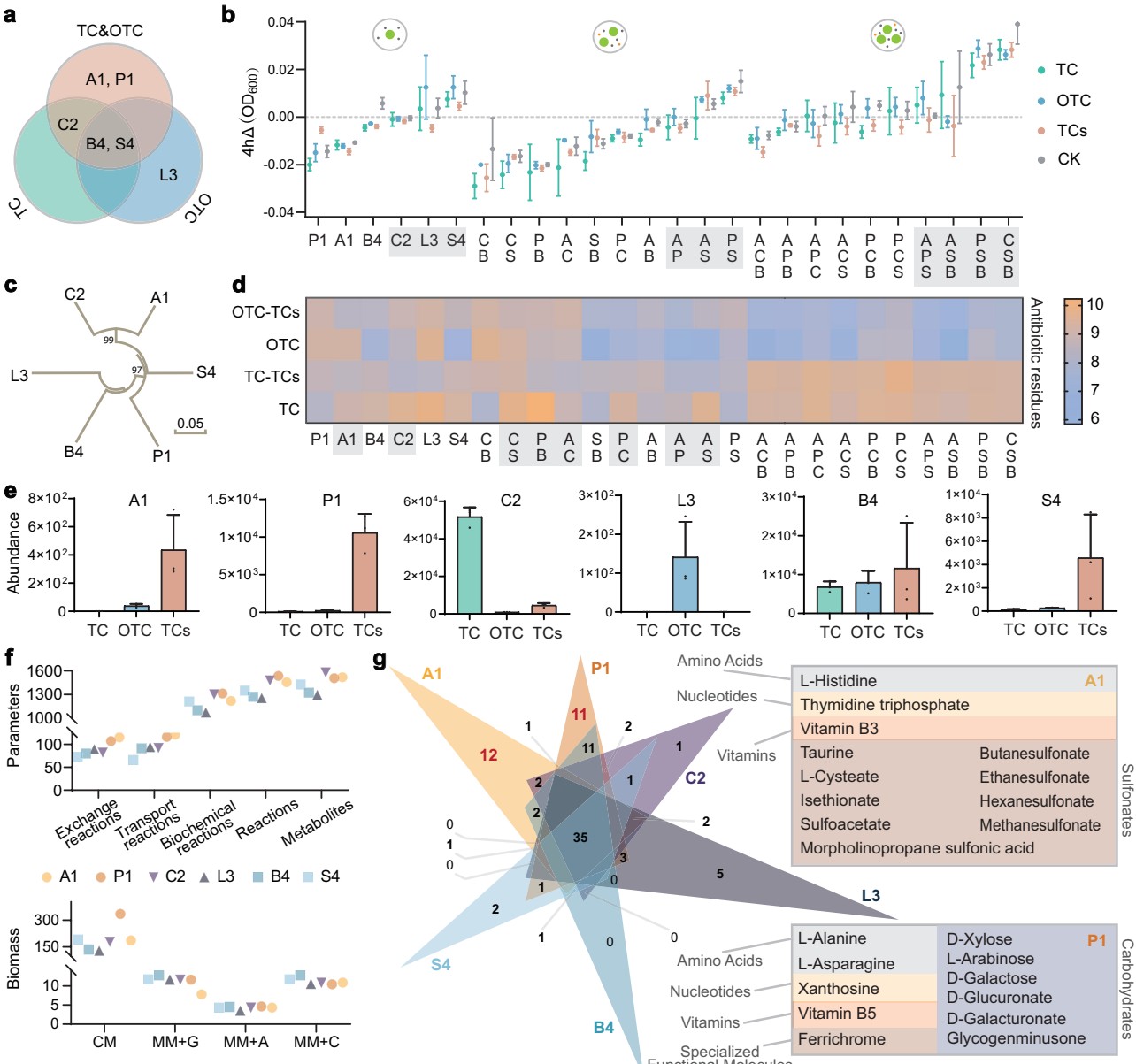

**Fig. 3 | Isolation and metabolic characterization of keystone species using GSMMs. a** Sources of the six isolated keystone species from TC-, OTC-, and TC&OTC-acclimated consortia. **b** Biomass production (ΔOD$_{600}$ after 4 h) of single-, co-, and tri-cultures under control (CK), single contamination (TC or OTC, 10 mg/L), and co-contamination (TCs, 10 mg/L each) conditions. Gray shading indicates increased biomass. Data are presented as mean values ± SD (*n* = 3 biological independent replicates). **c** Phylogenetic relationships of six keystone species constructed using the neighbor-joining method based on 16S rRNA sequences (1000 bootstrap replicates; values > 50% shown at nodes). Bar, 0.05 substitutions per nucleotide position. **d** Antibiotic removal efficiency of single-, co-, and tri-cultures as shown in (**b**), displaying average residual concentrations of TC and OTC in single

contamination, and both antibiotics in co-contamination. Gray shading indicates improved removal under co-contamination compared to single contamination. **e** Relative abundance of each keystone species within their original consortia. Data are presented as mean values ± SD (*n* = 3 biological independent replicates). **f** Model characteristics of the six genome-scale reconstructed models, with simulated biomass yields under four media conditions: complete medium (CM), MM +glucose, MM+citrate, and MM+acetate. **g** Venn diagram showing overlap of exchange reactions among strains, with unique metabolites for A1 and P1 listed separately. TC tetracycline, OTC oxytetracycline, TCs TC&OTC, MM mineral salts medium. Source data for this figure is available in the Source data file.

representative bacterial strains and 57 interspecies combinations under single or mixed pollutant conditions. Communities exposed to mixed pollutants exhibited stronger responses, including increased metabolite secretion and taxonomic diversity. The enhanced metabolic cross-talk enabled the secretion of diverse metabolites for stress resistance, consistent with studies highlighting the role of metabolic cross-feeding in community stability[19]. Notably, these key metabolites were secreted by taxa with stable abundance profiles, indicating their

foundational role in supporting the resilience of synthetic microbiome under complex contamination. This work elucidates secretion-based survival mechanisms in natural microbial communities under multi-pollution and provides a conceptual and technical foundation for the rational design of metabolic secretion-driven synthetic microbiomes for environmental bioremediation.

Understanding microbial interactions is essential for elucidating survival mechanisms under co-contamination[5,20]. Our study

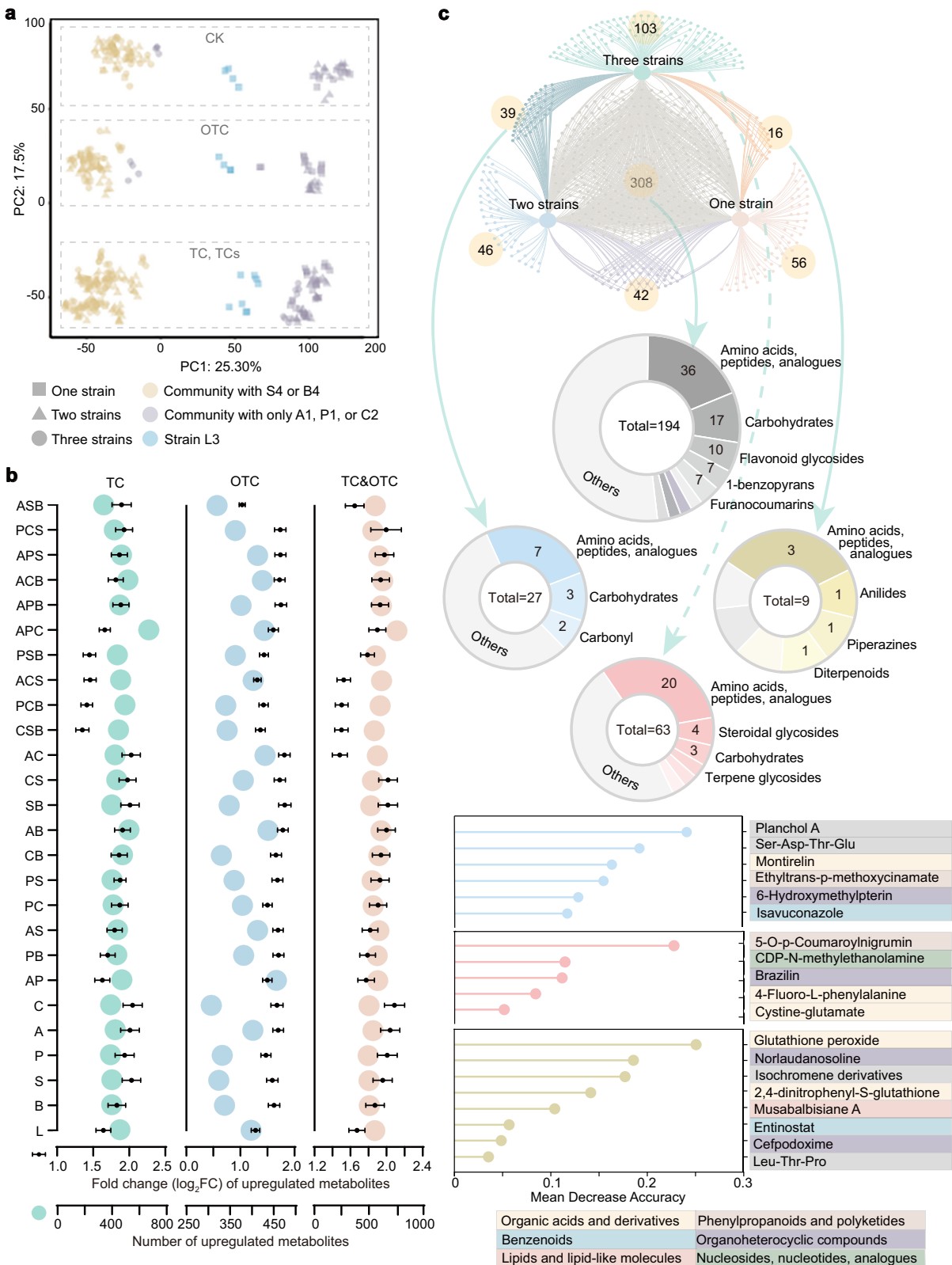

demonstrated that in TCs systems, both co- and tri-strain consortia exhibited higher metabolic secretion diversity than monocultures, with combination-specific exchange patterns. Notably, we identified unique cross-talk metabolites absent from simpler consortia—only half of which could be annotated—underscoring the largely untapped diversity of microbial metabolic networks. Consistent with previous simulations showing that AA and OA supplementation reduce sugar-

based competition, our results confirm that AA/OA addition in co-contaminated systems can facilitate carbon source utilization and modulate inter-strain competition. While nutrient limitation typically restricts microbial growth in undisturbed environments[21–23], co-contamination induces two survival strategies: enhanced production of cross-talk metabolites or upregulation of protective compounds. Metabolite profiling consistently identified amino acids and

**Fig. 4 | Identification of keystone metabolites through metabolomics analysis.** **a** Principal component analysis (PCA) of untargeted metabolomics data from single-, co-, and tri-strain communities under three treatment conditions: antibiotic-free control (CK), single-antibiotic treatments (TC or OTC), and combined-antibiotic treatment (TCs), delineated by dashed-line frames. **b** Quantitative analysis of significantly upregulated metabolites (compared to CK; Student's *t*-test, two-sided, $p < 0.05$, VIP ≥ 1 in OPLS-DA), displaying metabolite counts (represented by color-coded circles) and mean fold-change ($\log_2$FC ± SEM). The microbial consortium abbreviations (APCLSB) in the figure correspond to strains A1, P1, C2, L3, B4, and S4, respectively. Data are presented as mean ± SD; $n = 4$ biological replicates. **c** Functional characterization of metabolites under combined antibiotic stress (compared to CK; Student's *t*-test, two-sided, $p < 0.01$, VIP ≥ 2 in OPLS-DA). *Top:* Venn diagram and functional annotation of TCs-associated metabolites. Small nodes represent individual metabolites, while yellow and grey circles indicate

metabolites shared among or unique to different microbial communities, respectively. Connecting lines represent predicted metabolite production capability. Circular sectors display Human Metabolome Database (HMDB)-annotated metabolite classes, with central numbers indicating the total count of annotated metabolites. *Center:* Circular annotation plot specifically highlights metabolite distribution patterns. The grey ring displays metabolites common to all three community types; the yellow ring represents metabolites shared between single- and tri-strain communities; the blue ring denotes metabolites shared between dual- and tri-strain communities; the light red ring highlights metabolites unique to the tri-strain consortium. *Bottom:* Adjacent lollipop plots depict key metabolites predicted by random forest analysis, colored according to the corresponding categories they belong to (yellow/blue/light red). TC tetracycline, OTC oxytetracycline, TCs TC&OTC. Source data for this figure is available in the Source data file.

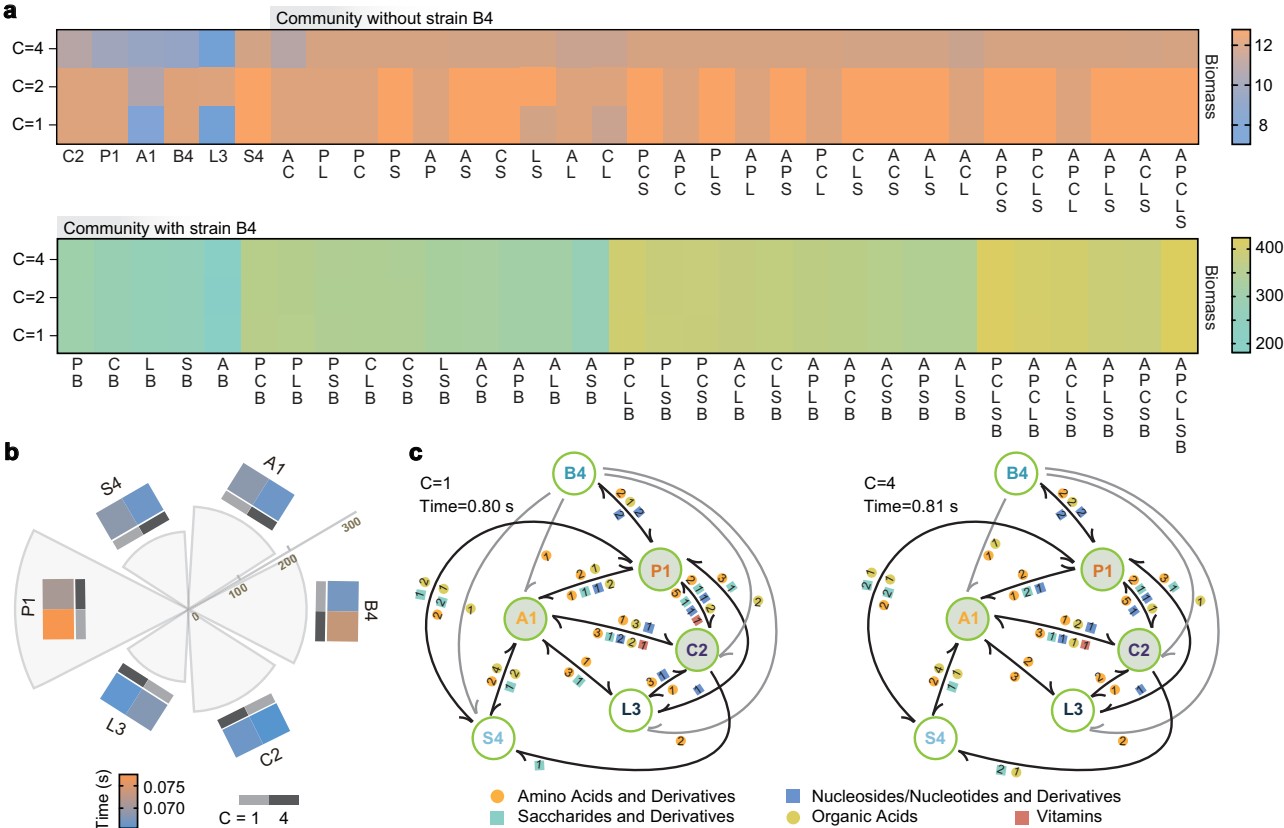

**Fig. 5 | Simulation of multiple environmental conditions and analysis of cross-feeding interactions in synthetic microbiomes. a** Simulated optimal biomass production under varying environmental conditions. Using the SuperCC modeling framework, the maximum biomass was simulated for single- to six-strain combinations under three nutritional scenarios: (1) MM supplemented with 100 mmol/gDW glucose (single carbon source, $C = 1$); (2) MM with 50 mmol/gDW each of glucose and citrate (dual carbon sources, C = 2); and (3) MM with 25 mmol/gDW each of glucose, citrate, acetate, and fumarate (quadruple carbon sources, $C = 4$). The top panel displays combinations excluding strain B4, while the bottom panel shows combinations including strain B4. **b** Strain-specific maximum biomass production and corresponding growth time under $C = 1$ and $C = 4$ conditions, assuming exclusive allocation of metabolic resources to each target strain. In the circular

diagram, sector size represents biomass, and segment color indicates growth time for each condition. **c** Flux balance analysis illustrating interspecies metabolic exchange under the objective of maximizing total community biomass. Each node (large circle) represents a bacterial strain, with directed arrows indicating the direction of metabolite transfer. Color-coded peripheral symbols (small circles) represent metabolite classes, and numeric labels indicate the specific number of metabolite types exchanged within each class. The annotation in the upper left corner shows the number of carbon sources available in the medium and the simulation time (20 h) required for the microbial community to reach optimal growth. Grey lines represent unidirectional exchange reactions originating exclusively from strain B4. Source data for this figure is available in the Source data file.

derivatives (AAs) as the most abundantly exchanged compounds (Fig. 5c), followed by OAs and carbohydrates. Although soil studies have established AAs as drivers of probiotic community development[24,25], our findings suggest a more context-dependent role in polluted environments. Specifically, while AAs were ubiquitously exchanged, they appear to regulate metabolic flux rather than serve as direct growth promoters. Notably, experimental validation further

showed that AA supplementation exerted minimal effects on growth compared to control treatments across four pollution scenarios, with growth inhibition observed under biocide and sulfonamide exposure (Fig. 6a). We propose a priming mechanism whereby AA exchange facilitates the synthesis and transfer of complex macromolecules, thus supporting the formation of stress-adapted metabolic networks in polluted environments.

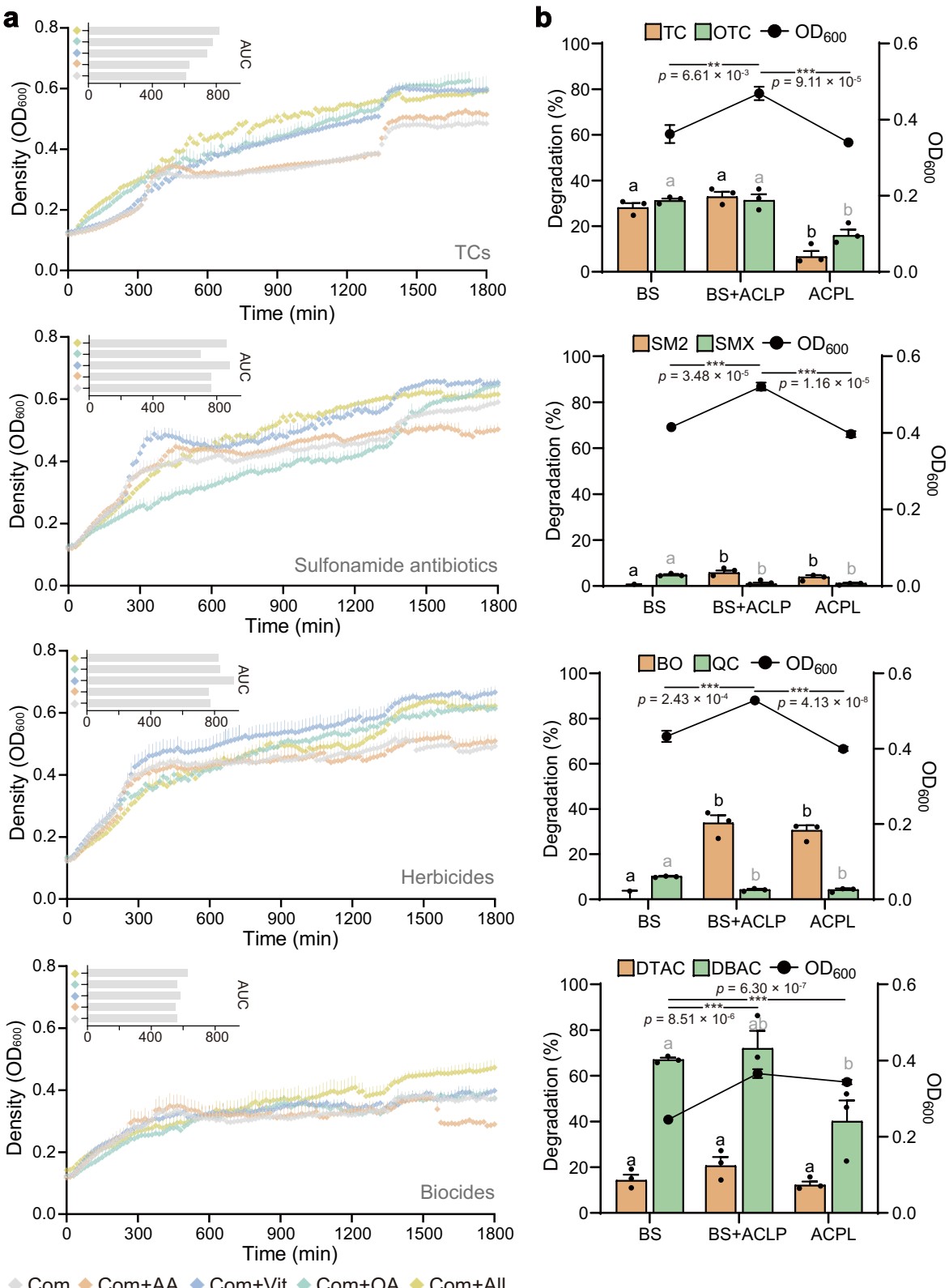

While microbial cross-feeding enables cooperative growth, we identified a paradox: why do strains A1 and P1 thrive exclusively in co-contaminated environments, whereas B4/S4 persist across diverse consortia? Metabolic modeling revealed that A1 and P1 possess more unique bidirectional metabolic exchange capabilities (Fig. 3g). Under simulated single-pollutant, nutrient-limited conditions, optimal growth simulations revealed intensive cross-feeding among A1/P1/C2

(Fig. 5c). However, natural microbial communities often exhibit net antagonistic interactions in such environments[2], which may explain why auxotrophic degraders like P1 and C2 fail to persist despite their metabolic potential. Paradoxically, auxotrophy becomes advantageous when essential metabolites are readily available[26], as evidenced by the enhanced survival of these strains in compound-polluted (multi-carbon) environments, where cross-feeding dependence is reduced.

**Fig. 6 | Experimental validation of predicted metabolite-mediated interactions under co-contaminated conditions. a** Growth promotion by metabolite supplementation. Growth curves illustrate the responses of microbial communities to four types of combined pollution (10 mg/L each), including tetracyclines (TC and OTC), sulfonamide antibiotics (SMX and SMT), herbicides (BO and QC), and biocides (DTAC and DBAC). Various metabolite combinations were added to the culture systems and monitored at 15-min intervals for 1800 min at 30 °C using a microplate reader, resulting in 9680 growth measurement data points per treatment. The control group (Com) was cultured in basal medium (containing 10% peptone), while the experimental groups were supplemented with additional amino acids (Com+AA: L-histidine, L-glutamate, L-phenylalanine, N-formyl-L-methionine, glycine, and L-proline), vitamins (Com+Vit: vitamin B1, vitamin D2, and niacin), organic acids (Com+OA: palmitate, myristic acid, and succinate), or a combination of all key metabolites (Com+All). The top-left inset bar plot quantifies the area under each growth curve. Data are presented as mean ± SD ($n = 4$ biologically independent replicates). **b** Validation of the altruistic effects of strains B4 and S4 (including promoting microbial community growth and pollutant removal). Bar plots represent the removal efficiencies of mixed pollutants by three microbial consortia (left $Y$-axis). Lowercase letters above the bars indicate statistically significant differences between groups (one-way ANOVA, $p < 0.05$), where black letters represent differences among pollutants in orange color blocks, and gray letters represent differences among pollutants corresponding to green color blocks. Line plots represent bacterial growth (measured as absorbance at $OD_{600}$; right $Y$-axis). $**p < 0.01$, $***p < 0.001$. The tested consortia were: BS (strains B4 and S4), BS + ACLP (strains B4, S4, A1, C2, L3, P1), and ACPL (strains A1, C2, L3, P1). Data are presented as mean ± SD ($n = 3$ biologically independent replicates for degradation, $n = 5$ biologically independent replicates for biomass). TC tetracycline, OTC oxytetracycline, DTAC dodecyl trimethyl ammonium chloride, DBAC dodecyldimethylbenzylammonium chloride, BO bromoxynil octanoate, Bxn bromoxynil (intermediate metabolite of BO), QC quinclorac, SMX sulfamethoxazole, SMT sulfamethazine. Data are presented as mean values ± SEM. Source data for this figure is available in the Source data file.

Interestingly, the top 10 keystone strains maintained minimal exchange reactions yet achieved higher growth rates, allowing them to dominate their ecological niches (Supplementary Fig. 9). Conversely, lower-abundance strains exhibited an increasing reliance on metabolic interactions (Fig. 2h). Beyond metabolic exchanges, our observations are consistent with a role of vitamins in microbial growth and metabolism[21,24,27]. In synthetic microbiomes, vitamin supplementation was associated with increased microbial biomass and enhanced pollutant removal. We also verified that the growth-promoting effect of metabolite supplementation stems from the specific metabolites themselves, rather than the additional carbon input (Supplementary Fig. 10). Strikingly, metabolic network analysis revealed thiamine exchange occurred exclusively among strains A1/P1/C2. This targeted vitamin-sharing mechanism, wherein strains release essential nutrients into the extracellular environment, likely underpins their stress resilience. Collectively, this metabolic autonomy-interdependence continuum represents a critical survival strategy for microbial communities facing compound pollution stress.

In contrast to the reduced microbial diversity observed under high-stress conditions like extreme weather or drought[28], organic pollutants can serve as supplementary carbon and nitrogen sources, whose degradation and utilization depend on microbial metabolic cooperation[29]. However, recent studies have shown that pollutant combinations containing OTC significantly disrupt microbial community structure[2], thereby challenging the assumption that increased pollutant loads necessarily promote microbial survival. Unlike previous studies using employing artificially combined single-strain assemblages[2,12], our results demonstrate that natural microbial communities exhibit more robust interactions, maintaining metabolic activity even under combined TCs stress through enhanced cross-feeding. These findings not only deepen our understanding of microbial adaptation strategies under co-contamination conditions but also underscore the importance of indigenous strains in remediating antibiotic-contaminated environments. Building on our prior work, we have successfully developed high-efficiency synthetic microbiomes capable of herbicide degradation in polluted soils[5]. Most current synthetic consortium designs adopt the Degrader&Helper model, identifying helper strains through correlation-based analysis with target degraders (Table 1). For instance, the A1&P1&C2 consortium characterized in this study exhibits tightly coordinated interactions mediated by unique cross-talk metabolites that enhance metabolic function.

Microbial populations are fundamentally constrained by resource availability, as limiting micronutrients such as C/N/P constrain community size and drive interspecies competition[27]. Our findings highlight the critical role of microbial interactions in optimizing microbiome functionality[30,31], particularly through the presence of altruistic potentiator strains acting that serve as the community's silent guardians. These strains bridge defense and offense by supporting Degrader&Helper strains during biodegradation processes. Strains B4 and S4 serve as archetypes of this ecological function[32], providing complementary metabolic support. These ubiquitous, high-abundance strains exhibit extensive overlap in exchange/transport reactions with other community members (Fig. 3g)—a metabolic redundancy known to enhance ecosystem stability, diversity, and competitive adaptability[13,33]. This support is particularly evident through the increased production of nitrogenous metabolites, including free AAs and diverse peptide compounds. Notably, B4/S4-containing consortia produced fewer unique metabolites overall but demonstrated significant enrichment in AA and peptide diversity relative to consortia lacking these strains (Supplementary Fig. 6). Experimental validation confirmed that Degrader&Helper consortia supported by B4 and S4 achieved higher growth and greater metabolic activity (Fig. 6b). Furthermore, the potentiator strain B4 exhibits uniqueness: we tested species from the same genus or family, yet no such potentiating effect was observed (Supplementary Fig. 11). This metabolic profile reflects an evolved cooperative strategy for nutrient acquisition under resource-limited conditions. Remarkably, despite exhibiting slower growth and a competitive disadvantage in multi-substrate environments, B4 consistently improved overall community performance by donating metabolites to neighboring strains. These slow-growing yet environmentally resilient microorganisms, often overlooked due to their consistent abundance, function as critical ecosystem potentiators (Table 1). They significantly contribute to microbial community structure and facilitate successful colonization, underscoring their essential role in future synthetic microbiome design.

Microorganisms are pivotal to global biogeochemical cycles[34,35], particularly those capable of degrading pollutants, which play a critical role in environmental remediation of contaminated water and soil systems[5,29]. While recent advancements have produced engineered strains capable of simultaneously degrading multiple organic pollutants, the rapid emergence of novel recalcitrant contaminants—such as microplastics and other persistent organic pollutants—presents new challenges, as their recent emergence in the environment has not allowed sufficient time for natural microbial adaptation[3,36]. We propose the construction of synthetic microbiomes incorporating native potentiator strains to regulate the metabolic vitality of microbial communities through key metabolites, thereby supporting the formation of stable systems that can sustain native pollutant degraders. The GSMM-based SuperCC framework enables the identification of key potentiator strains whose metabolic redundancy enhances community resilience in both soil and aquatic environments. Our study successfully developed a multifunctional DHP-Com consortium derived from co-contaminated water, which demonstrated cross-environmental adaptability for composite pollutant remediation.

However, microbial communities are dynamic systems undergoing continuous succession[37], and our focus on day-30-adapted populations suggests that transitional community dynamics may have been overlooked. This consortium design framework warrants further validation and adaptation for soil remediation applications, offering a potentially versatile strategy for addressing a wide range of environmental contamination scenarios.

## Methods

### Enrichment of TCs-degrading consortia

Activated sludge was collected from the aeration tank of a municipal wastewater treatment plant (23°40′N, 113°22′E) in Huadu District, Guangzhou, China. Enrichment cultures were established in 500-mL aluminium foil-wrapped conical flasks containing 200 mL of sterile enrichment medium (EM) and 100 mL raw sludge (see Supplementary Method 1). Three experimental groups were designed: (i) EM supplemented with 5 mg/L tetracycline (TC group), (ii) EM supplemented with 5 mg/L oxytetracycline (OTC group), and (iii) EM simultaneously supplemented with 5 mg/L tetracycline and 5 mg/L oxytetracycline (TC&OTC group, TCs). All flasks were incubated aerobically at 30 °C with continuous shaking at 150 rpm under dark conditions. After 7 days of initial acclimatization, 100 mL of microbial suspension from each group was aseptically transferred to fresh EM containing 10 mg/L of the respective antibiotic(s), while maintaining the same cultivation conditions. The antibiotic concentration was progressively increased in 5 mg/L increments via serial subculturing every 7 days. This acclimatization process continued for 10 cycles, with three biological replicates per treatment. Samples from the final round of enrichment for each treatment group were collected and stored at −80 °C for DNA extraction and amplicon sequencing. A 5% (v/v) inoculum from each enriched consortium was subsequently transferred into degradation medium (DM) containing 50 mg/L TC or 25 mg/L OTC and incubated for 0, 3, 7, and 10 days, and analyzed by LC-MS to quantify residual antibiotic concentrations (see Supplementary Method 2 for detailed protocol).

### Bacterial 16S rRNA gene amplicon sequencing and analysis

Total DNA was extracted using the ALFA Soil DNA Extraction Kit. DNA integrity was assessed by 1% agarose gel electrophoresis, while concentration and purity were measured using a NanoDrop One. The V3-V4 hypervariable regions of bacterial 16S rRNA genes were amplified using barcoded universal primers 338 F (5′-ACTCCTACGGGAGGCAGCA-3′) and 806 R (5′-GGACTACHVGGGTWTCTAAT-3′). Sequencing libraries were constructed according to the standardized protocol of the NEBNext® UltraTM II DNA Library Prep Kit for Illumina®. Finally, the amplified products were sequenced using the Illumina NovaSeq 6000 platform with PE250 at Guangdong Magigene Biotechnology Co., Ltd., Guangzhou, China. The sequencing data were analyzed using the QIIME2 pipeline (v2020.11.0)[38] with the DADA2[39] plugin for primer removal, denoising, filtering, merging, and chimera removal, generating high-resolution amplicon sequence variants (ASVs) with 100% sequence identity. Taxonomic classification was performed using Usearch-sintax (v10.0.240) against the SILVA SSU 138 database[40], with a confidence threshold of 0.8 applied for taxonomic assignments. $\alpha$-diversity indices were calculated using Usearch-alpha_div (v10.0.240). Functional profiling was conducted using PICRUSt2[41]. The Greengene IDs associated to each ASV were systematically aligned against the KEGG database[42].

### Keystone isolation and whole genome sequencing

All three enriched cultures underwent multiple rounds of serial dilution and plating on Luria-Bertani (LB) agar. Individual colonies were isolated and further cultivated for taxonomic characterization before preservation in 15% (v/v) glycerol at −80 °C. The 16S rRNA gene was amplified using universal primers 27F (5′-AGAGTTTGATCMTGGCTCAG-3′) and 1492R (5′-TACGGYTACCTTGTTACGACTT-3′). Strain classification was performed using the EzBioCloud server, with phylogenetic analysis performed using the neighbor-joining method in MEGA 11.0[43]. Six predominant bacterial genera were identified, from which one representative strain per genus was selected for subsequent modeling and validation experiments. Whole-genome sequencing of the six strains was performed using both Nanopore (third-generation) sequencing and Illumina second-generation sequencing platforms at Wuhan Bena Biotechnology Service Co., Ltd (Wuhan, China). For comprehensive functional annotation, the genomes were analyzed against multiple databases, including NCBI non-redundant (NR), SwissProt, KEGG, and COG using blastp.

### Metabolomic profiling of key species combinations

This study systematically evaluated bacterial combinations including the following: (1) six single strains (A1, P1, C2, L3, S4, B4), (2) ten pairwise combinations (AP, AC, AS, AB, PC, PS, PB, CS, SB, CB), and (3) ten three-strain consortia (APC, APS, APB, ACS, ACB, ASB, PCS, PCB, PSB, CSB). Individual strains were cultured to the mid-logarithmic phase in lysogeny broth medium (LB), harvested by centrifugation, washed twice with minimal medium (MM), and standardized to $OD_{600} = 1.0$. Consortium inoculations were conducted under four treatment conditions: (i) MM + 10 mg/L TC, (ii) MM + 10 mg/L OTC, (iii) MM + 10 mg/L TC + 10 mg/L OTC, and (iv) antibiotic-free MM control. All cultures were maintained at consistent inoculum sizes with four biological replicates and incubated at 30 °C with shaking at 180 rpm. Growth kinetics were monitored via $OD_{600}$ measurements at 0, 2, 3, and 4 h, with parallel sampling for antibiotic quantification.

Following 4-hour cultivation, a total of 416 cultures were centrifuged ($6000 \times g$, 10 min), and supernatants were flash-frozen and lyophilized. Metabolites were extracted and identified at Majorbio Biotechnology Co., Ltd. (Shanghai, China) (detailed protocol in Supplementary Method 3). Variables with a relative standard deviation (RSD) > 30% in QC samples were excluded, and the data were $log_{10}$-transformed to produce the final data matrix. Model stability was assessed using sevenfold cross-validation.

### Construction of single- and multi-species models

Single-strain metabolic models were constructed for six bacterial genomes by first generating draft models from protein sequences using ModelSEED[44], followed by refinement in MATLAB using the COBRAToolbox (v3.0) to ensure positive biomass flux under open exchange condition[45]. Missing reactions were manually supplemented based on literature and annotations from phylogenetically related species, with additional gap-filling performed using the KEGG, UniProt, BiGG, IMG, and MetaCyc databases[46]. Only reactions supported by genetic evidence in the genome of the target strain or its closely related species were retained. All reactions were standardized for consistency[46,47]. Through iterative modification, model functionality was validated in MM containing a single carbon source (glucose, citrate, or acetate), during which we unified database-specific reaction IDs, corrected elementally imbalanced reactions, and eliminated futile cycles. By reviewing the literature, we modified the direction of reversible reactions and the mass-charge balance of metabolic reactions, and modified reactions utilizing $CO_2$ and reversible $CO_2$ reactions into unidirectional reactions. The finalized models successfully predicted biomass production across alternative C/N sources, in agreement with experimental observations (Supplementary Data 5). For community simulations covering single-, co-, and tri-cultures, the SuperCC framework[5] was employed to model growth dynamics under two nutritional conditions: (1) a single carbon source (100 mmol/gDW glucose) and (2) mixed carbon sources (25 mmol/gDW each of glucose, citrate, acetate, and fumarate). Except for the essential nitrogen source

($NH_4^+$) which was set to 100 mmol/gDW, the contents of all other inorganic substances without carbon (C) or nitrogen (N) were set to 1000 mmol/gDW. To evaluate the robustness of our results to variations in the four-carbon mix (glucose, citrate, acetate, fumarate), we performed sensitivity analyses; results confirming robustness are presented in Supplementary Tables 4 (experimental validation) and 5 (modeling validation). The relative change in biomass remained below 10% under all perturbations. These findings confirm that our results are robust to variations in the specific four-carbon mix used.

Additionally, single-strain metabolic models were developed for the Top-50 species in each of the three consortia using genome sequences from NCBI and JGI, resulting in a total of 70 species (Supplementary Data 5). Following our established pipeline, biomass production and growth time were predicted for each strain in complete medium (Supplementary Table 1). Potential metabolic interactions were analyzed through co-occurrence networks generated with R/ igraph and visualized in Gephi (Version 0.9.2) using Fruchterman–Reingold layout[48].

Flux distributions were computed using parsimonious flux balance analysis (pFBA)[49] to maximize biomass production while minimizing total nutrient uptake fluxes[5]. The mathematical framework of model construction is shown below. All species in the community are represented as $K$. For each species $k \in K$, the FBA for predicting maximum growth is formulated to maximize the biomass flux $v_{biomass}^k$, subject to:

$$\sum S_{mn}^k v_n^k = 0, \forall_m \in M^k, \forall_n \in N^k \tag{1}$$

$$LB_n^k \leq v_n^k \leq UB_n^k, \forall_n \in N^k \tag{2}$$

$S_{mn}^k$ is the stoichiometry for metabolite $m$ in reaction $n$ for species $k$. The reaction flux $v_n^k$ is measured in mmol gDW$^{-1}$h$^{-1}$ (general reactions) or h$^{-1}$ (biomass reaction). All metabolites $m$ and reaction $n$ of species $k$ belong to the metabolite set $M^k$ and reaction set $N^k$, respectively. The flux range of each reaction is constrained by its lower bound $LB_n^k$ (minimum uptake flux) and upper bound $UB_n^k$ (maximum secretion flux).

The mass balance of secretions and uptakes of each species in the extracellular space in the microbial community is stated as follows:

$$\left( \sum_{k \in K} V[ex]_m^k \right) + IP_m^k - EP_m^k = 0, \forall_m \in M^{com} \tag{3}$$

$$max \sum_{k \in K} c^k V_{biomass}^k \tag{4}$$

$v[ex]_m^k$ denotes the exchange reaction flux for metabolite $m$. The community import and export rates of metabolite $m$ are represented by $IP_m^k$ and $EP_m^k$, respectively. The set of shared metabolites in the community is denoted as $M^{com}$. The objective function of the community model is defined as the weighted sum of biomass fluxes $v_{biomass}^k$ from all organisms (Equation [4]), where the weighting coefficients are specified by vector $c^k$. Four commonly used scenarios were provided, including (1) equal abundance for each organism; (2) no limitation for any organisms, meaning that the biomass of each organism is allowed to be zero; (3) defining biomass of a specific organism as community biomass (used for identifying organisms in a community that could improve the growth of the target organism); and (4) any defined abundances.

## Verification of key metabolites' role on co-contamination remediation

Individual strains A1, P1, C2, L3, S4, and B4 were cultivated to the logarithmic growth phase in LB. Cells were collected by centrifugation (6000 × $g$, 10 min) and resuspended in fresh medium, and adjusted to an OD$_{600}$ of 1.0. Equal volumes of each strain suspension were mixed to form a synthetic microbiome. The consortium was inoculated into MM containing 10% (w/v) peptone along with four distinct co-contaminated mixtures: (i) TCs (10 mg/L TC and 10 mg/L OTC), (ii) sulfonamide antibiotics (10 mg/L SMX and 10 mg/L SMT), (iii) herbicides (10 mg/L BO and 10 mg/L QC), and (iv) biocides (10 mg/L DTAC and 10 mg/L DBAC). Five experimental treatments were established for each pollutant mixture: (1) Com: Microbial consortium + 10 mg/L two pollutants; (2) Com+AA: Com + 10 mg/L amino acids (L-histidine, L-glutamate, L-phenylalanine, N-formyl-L-methionine, glycine, and L-proline); (3) Com+Vit: Com + 10 mg/L vitamins (vitamin B1, vitamin D2, and niacin); (4) Com+OA: Com + 10 mg/L organic acids (palmitate, myristic acid, and succinate); (5) Com+All: Com + all key metabolites (10 mg/L each). 200 μL aliquots from each treatment were transferred to 96-well plates ($n$ = 4 replicates) and monitored at 15-min intervals for 1800 min at 30 °C using a microplate reader, resulting in 9680 growth measurements per treatment. Parallel batch cultures were incubated aerobically at 30 °C with shaking (180 rpm) for 24 h, followed by quantification of residual pollutants ($n$ = 3 biological replicates).

## Degradation capacity of six keystone strains against TC and OTC

Strains A1, P1, C2, L3, S4, and B4 were grown to the logarithmic phase in LB, harvested (6000 × $g$, 10 min), washed, and resuspended in fresh medium to an OD$_{600}$ of 1.0. Degradation assays were conducted in MM-P medium containing 30 mg/L TC or OTC and incubated for 3 days. Each strain was tested in triplicate, and uninoculated controls were included to account for the natural degradation of TC and OTC.

## Equivalent glucose supplementation experiments

Control experiments were conducted to determine whether the observed growth-promoting effects of metabolite supplementation were attributable to additional carbon input rather than to the specific metabolites themselves. A microbial consortium composed of strains A1, P1, C2, S4, B4, and L3 was inoculated into MM-10% P medium containing 10 ppm TC and 10 ppm OTC.

For Treatment 1, amino acids including L-proline, N-formyl-L-methionine, L-glutamic acid, L-phenylalanine, glycine, and L-histidine (10 ppm each) were added, corresponding to a total carbon molar amount of $4.62128 \times 10^{-5}$ mol C. In Treatment 2, glucose was supplemented to provide an equivalent total carbon molar amount as in Treatment 1, while in Treatment 3, glucose was added to double this carbon molar amount. Similarly, for the vitamin treatment (10 ppm each of vitamins B$_3$, B$_1$, and D$_2$; total $3.18438 \times 10^{-5}$ mol C) and the organic acid (OA) treatment (10 ppm each of palmitic acid, tetradecanoic acid, and succinic acid; total $5.25235 \times 10^{-6}$ mol C), corresponding glucose controls were prepared by supplementing glucose at equivalent or double molar amounts of carbon relative to the added metabolites.

Microbial growth was monitored by measuring the optical density at 600 nm (OD$_{600}$) over a 2400-min incubation period. Each treatment was performed in quadruplicate ($n$ = 4 biologically independent replicates).

## Validation of altruistic effects via enhancer strain (B4 and S4) omission

Three microbial consortia (BS: B4 and S4; BS + ACLP: B4, S4, A1, C2, L3, P1; ACLP: A1, C2, L3, P1) were inoculated into MM-P medium containing four sets of mixed pollutants (10 ppm each): (1) TC + OTC; (2) DTAC + DBAC; (3) BO + QC; (4) SMX + SMT. Bacterial growth was measured at OD$_{600}$, and pollutant removal was quantified.

Experiments were performed with independent biological replicates ($n = 5$ for biomass, $n = 3$ for degradation). One-way ANOVA was used for statistical analysis; lowercase letters indicate significant differences ($p < 0.05$).

## Community member substitution test

Different bacterial consortia were inoculated into MM-P medium containing 10 ppm TC and OTC and incubated for 27 h. The tested consortia were: CP (*Comamonas* sp. C2 + *Providencia* sp. P1), CPB (C2 + P1 + *Brevundimonas* sp. B4), CPB-1 (C2 + P1 + *Brevundimonas vesicularis*, same genus as B4 but different species), and CPB-2 (C2 + P1 + *Caulobacter vibrioides*, same family as B4 but different genus). Bacterial growth was monitored by optical density at $OD_{600}$, and TC degradation was quantified after 27 h. Data are presented as mean ± SD of four independent biological replicates.

## Statistics and reproducibility

Statistical analyses and data visualization were performed using packages in the statistical program R v4.0.3. The statistical significance of $\alpha$-diversity differences was evaluated using one-way ANOVA. $\beta$-diversity was computed using Bray-Curtis dissimilarity metrics and visualized using Principal Coordinates Analysis (PCoA) via the vegan package (v2.5-7). Principal component analysis (PCA) and orthogonal partial least squares discriminant analysis (OPLS-DA) were performed using the R package ropls (Version 1.6.2). Metabolites with a variable importance in projection (VIP) ≥ 1 and a $p$-value < 0.05 (Student's $t$-test) were considered significantly different metabolites[50]. The study design and statistical analyses were conducted to ensure the rigor and reproducibility of the results. No statistical method was used to predetermine sample size. The experiments were not randomized.

## Reporting summary

Further information on research design is available in the Nature Portfolio Reporting Summary linked to this article.

## Data availability

All amplicon sequencing data have been deposited in the NCBI Sequence Read Archive (SRA) under the accession number PRJNA1153784 [https://www.ncbi.nlm.nih.gov/bioproject/1153784]. The GenBank accession numbers for genome sequences of the isolates are CP190002-CP190008 under the accession number PRJNA1240688. The metabolomics data have been deposited in the MetaboLights database under the accession number MTBLS12809. The data that support this study are available within the article and its Supplementary Information files. Source data are provided with this paper. The optimization models, media, and SuperCC function used in MATLAB are available at https://github.com/ruanzhepu/superCC.git. Source data are provided with this paper.

## Code availability

The code used for the analysis is available on GitHub (https://github.com/ruanzhepu/superCC)[51].

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

## Acknowledgements

This study was supported by the National Key Research and Development Program of China (2023YFC3905800 [R.Q. and Z.R.], 2023YFC3707600 [Z.R.]), the National Natural Science Foundation of China (42307006 [Z.R.]), Key Realm Research and Development Program of Guangdong Province (2023B0202020001 [R.Q.]), Guangdong Provincial Science and Technology Plan Project (2021B1212040008 [R.Q.]).

## Author contributions

Z.R., J.T., and R.Q. conceived the overall study design. J.T., Q.F., K.Y., and D.L. performed the experiments. Z.R. and J.T. carried out bioinformatics analysis. Z.R. reconstructed models and performed modeling analysis. R.Q. supervised the research. Z.R. and J.T. wrote the manuscript. Y.C., P.W., Z.N., J.C., and R.Q. revised the manuscript. Z.R. and R.Q. acquired the funding. All authors read and approved the final manuscript.

## Competing interests

The authors declare no competing interests.
