## [Transparent Peer Review file · Nature Communications]

Potentiators empower synthetic microbiomes as silent guardians against co-contamination

Corresponding Author: Professor Rongliang Qiu

Version 0:

Reviewer comments:

Reviewer #1

(Remarks to the Author)

GENERAL COMMENT

This manuscript by Ruan et al. presents an ambitious and comprehensive study integrating experimental microbiome acclimation, multi-omics analyses, and genome-scale metabolic modeling (GSMM) via the SuperCC framework to rationally design synthetic microbiomes optimized for complex pollutant remediation. The authors propose the novel “Degradation–Helper–Potentiator” (DHP-Com) paradigm, validate it with multi-strain culture experiments, untargeted metabolomics, and multi-pollutant degradation assays, and link their findings to mechanistic modeling predictions.

Noteworthy results: The work identifies keystone “potentiator” strains that enhance pollutant removal and community resilience across multiple contaminant classes, demonstrates that co-contamination can foster greater microbial diversity and metabolic cross-feeding, and introduces SuperCC as a modeling framework capable of simulating realistic community dynamics under multi-pollutant conditions.

Significance: The study addresses an important gap in synthetic microbiome design for real-world co-contamination scenarios, rather than idealized single-pollutant setups, and is likely to be of interest to researchers in environmental biotechnology, microbial ecology, and systems/synthetic biology.

Relation to established literature: The DHP-Com paradigm extends prior “Degradation–Helper” concepts (e.g., refs. 13–14, 15) by explicitly incorporating potentiator functions identified via integrated modeling and metabolomics. The work builds on, but clearly advances, existing studies such as Smith et al. 2024 and Ruan et al. 2024.

Support for conclusions: Overall, the experimental results and modeling outputs support the major conclusions. However, some claims, in particular, regarding potentiator roles, are inferred rather than directly demonstrated and would benefit from additional validation or caveats (see Major Comments 8–9).

Methodological soundness: The experimental design is diverse and generally robust, but certain aspects of the modeling approach (e.g., GEM reconstruction choices, simulation assumptions, parameter sensitivity) require clearer justification. The core methodology is sound for the field, but improved transparency would strengthen reproducibility and trust in the results.

Detail for reproducibility: The Methods section is extensive, but some parameters (exchange bounds, biomass coefficients, gap-filling criteria, metabolite identification thresholds) should be explicitly stated to fully meet reproducibility standards.

MAJOR COMMENTS

1. Model Transparency and Assumptions

Lines 431–472

How were the uptake bounds (LB) and secretion bounds (UB) for each metabolite determined for single-strain and community models? Were they uniform across all strains or strain-specific, based on annotation? Please clarify how uptake (LB) and secretion (UB) bounds were determined.

Did you use any thermodynamic feasibility filtering (e.g., TFA, min–max driving force) to remove energetically infeasible flux cycles? If not, discuss whether including TFA would alter predictions of cross-feeding patterns.

For exchange metabolite inclusion, what criteria were used (e.g., detected in metabolomics, KEGG annotation, gap-filling output)? Were dead-end metabolites removed from the community network?

How were biomass functions standardized and strain-specific precursors handled?

2. Community Model Configuration

Lines 453–472

Were biomass weighting coefficients (c_k in Eq. [4]) equal for all species, or did you use empirically measured relative

abundances to weight objectives?

3. Simulation Scenarios and Parameter Sensitivity

Lines 191–215

In the C=1 vs C=4 scenarios, did you hold total carbon molar input constant or increase total available carbon? If the latter, could increased total resource availability (not diversity) drive biomass gains?

Did you perform sensitivity analysis on carbon source composition to see if results were robust to the specific four-carbon mix used (glucose, citrate, acetate, fumarate)?

For the network visualizations in Fig. 5c, how was the “number of exchanged compounds” thresholded? Were only flux-carrying exchanges included, or all feasible exchanges under optimal growth?

4. Validation of Model Predictions

Lines 104–117

Were biomass predictions validated against experimental growth rate measurements for any subset of strains? If not, can you provide at least qualitative agreement with rank-order abundances?

For the observed “fast-growing, low-biomass strains” in co-contamination (line 116), could this be an artifact of model biomass equation stoichiometry?

5. Flux Predictions and Interpretation

Lines 205–215

Did you differentiate between obligatory vs facultative exchanges in the flux predictions? For example, were some metabolites exchanged only under C=1 but not under C=4?

The identification of B4 as an “altruistic keystone” (line 213) is compelling — did you test whether this classification holds under different objective functions (e.g., community-wide substrate uptake minimization)?

6. General Modeling Methodology

Lines 70–82

• The introduction describes the novelty of applying GSMM to co-contamination, but doesn’t discuss possible pitfalls (e.g., lack of spatial constraints, dynamic succession). In addition, it would be beneficial to know what the most suitable constraint-based modeling approaches are to use under these types of scenarios. Would you add 1–2 sentences acknowledging these modeling limitations and some modeling preludes to this work?

7. Experimental Design Controls

Lines 217–239, 474–490

Did you run abiotic controls (pollutants in medium without microbes) to measure any chemical degradation independent of biological activity? If so, where are these data shown?

Were non-acclimated consortia tested under the same pollutant stress to isolate the effect of the acclimation process?

In the metabolite supplementation experiments, could growth effects be due simply to extra C/N input rather than specific metabolites?

8. Statistical Robustness

Lines 410–429, 162–189

Were multiple testing corrections applied to metabolomics p-values? If so, which method? If not, would you consider applying FDR correction given the high number of features?

How many biological replicates (n) were used for each metabolomics condition? It would be helpful to state this explicitly in the figure legend for Fig. 4.

9. Mechanistic Evidence for Potentiators

Lines 191–215, 316–336

Are there any partial removal experiments or mutant strain data for B4 and S4 that could directly test the potentiator role?

Would you agree that potentiator function in your study is inferred rather than experimentally proven? Please add a cautionary note, making this explicit.

10. Choice of GEM Reconstruction Tool

Lines 431–444

Provide rationale for using ModelSEED over alternatives (CarveMe, gapseq, AGORA), detail manual curation steps, gap-filling criteria, and whether benchmarking against existing GEMs was performed.

Additional Reproducibility and Data/Code Availability

Please indicate where the data and code used in this study — including the SuperCC modeling scripts, genome-scale metabolic models, simulation parameters, and metabolomics datasets — can be accessed to reproduce the modeling and simulation results and to retrieve the processed and raw metabolomics data. If available, provide persistent links (e.g., GitHub, Zenodo, Figshare) and accession numbers for public repositories, along with any instructions necessary for execution.

MINOR COMMENTS

- Define “potentiator” in Abstract for broader readership.
- Clarify biomass coefficient weighting (line 470).
- Improve figure readability (Figs. 2i, 4, 5c, 6) and include complete metabolite class legends.
- Standardize “co-contamination” vs “co-pollutant” terminology.
- Include full media compositions in Supplementary.

Recommendation: Strong and methodologically advanced manuscript. However, I recommend major revisions to this manuscript to address several shortcomings. After a major overhaul, the manuscript will be significantly improved.

(Remarks on code availability)

Reviewer #2

(Remarks to the Author)

In this manuscript, the authors use a structured approach for targeted microbiome application by integrating top-down and bottom-up methods. This approach involves domesticating bacterial communities from natural water samples and leveraging in-silico strategies to optimize both the functional performance and stability of the resulting synthetic microbiomes. They design different treatment groups: adding a single antibiotic (either tetracycline or oxytetracycline) or co-adding both antibiotics. These treatments aim to analyze and identify keystone species in water samples, as well as enable directional enrichment of microbial communities in multi-pollutant contaminated environments. The authors further employ significance analysis to confirm these keystone species and use a compartmentalized community SuperCC approach—a powerful synthetic microbial community design framework independently developed by the study's authors—to evaluate metabolic flux exchange among different members of the microbial community.

The authors' study is quite interesting: they isolated six keystone species from polluted water, and all of these species align perfectly with the three functional roles proposed by the authors. Unlike the traditional synthetic microbial community model of "degrader bacteria + helper bacteria," the authors proposed a third bacterial type termed "Potentiators" and highlighted their pivotal role within the microbial community. The discovery and proposal of Potentiators have addressed the long-standing challenge in practical applications where microbial communities exhibit poor colonization and high mortality. These previously overlooked, inconspicuous bacteria play a crucial role in assisting functional microbial communities in colonization and degradation processes. To validate this hypothesis, the authors used the SuperCC genome-scale metabolic model to make predictions. By using different types and quantities of carbon sources to simulate single and composite pollution scenarios, they have significantly overcome the predicament where metabolic model simulations are hindered by the difficulty in clarifying microbial degradation pathways of pollutants in environments with emerging or composite pollutants. This approach offers theoretical and technical support for future simulations of microbial community interactions in complex polluted environments.

For the proposed DHP-Com framework, the authors further selected 8 organic pollutants belonging to 4 categories. Through this approach, they successfully enabled the microbial consortium to achieve efficient degradation capacity, as well as enhanced tolerance and growth capabilities. This work paves the way for broad application prospects in enhancing the adaptability and functionality of synthetic microbial communities across diverse complex environments. Overall, this is a potentially impactful paper.

Smaller comments:

- The format of references needs to be further confirmed. In some references, the species names in the titles are not italicized.
- All sequencing data in the manuscript (including raw data from metabolomic analyses) should be uploaded to public database platforms.
- Lines 465-466, Fix the line-breaking format of the formula.
- The formatting of P-values in the manuscript appears inconsistent. Additionally, a space should be inserted between "P" and "<". Please recheck and correct this throughout the entire manuscript.
- Lines 49-50 and 64-65 lack literature support. Please add relevant references and conduct a full-manuscript check.

Comments on Figures:

- Figure 1 could be appropriately enlarged, especially Figure 1a. As it presents the experimental framework and new concepts, the current size of the figure is somewhat small, which may affect readability.
- The line spacing between each row of the random forest results below Figure 4C is inconsistent.
- It is recommended to rename the y-axis label of Figure 6b to "degradation rate (%)".

(Remarks on code availability)

Reviewer #3

(Remarks to the Author)

SUMMARY

I found the manuscript intriguing and potentially impactful, but I had to re-read multiple sections due to organizational and definitional ambiguity. The scientific core feels strongest as a methodological advance—extending multi-strain GSMM/FBA to guide community design, while the specific strain findings are illustrative rather than the central contribution. Clarity around role definitions, statistical support for several comparative claims, and figure/text alignment would materially improve readability and confidence.

NOTEWORTHY RESULTS

- Model and bench alignment: Multistrain GSMM/FBA predictions are borne out experimentally, supporting the feasibility of model-guided, role-based consortium design.
- Mechanistic thread (vitamin rescue): Supplementation patterns (namely vitamins) directionally support a cross-feeding mechanism consistent with omics signals.

- Field impact: The work strengthens a known ecological principle (multi-member robustness) but advances the analysis toolkit for environmental biotech and microbial ecology by making >2-member modeling actionable.

SIGNIFICANCE

The ecological idea (in essence, that more species of the "potentiator" class can translate to greater performance/robustness, as I understood it) is not wholly novel per se, but formalizing it with genome-based multistrain modeling and using that to nominate compositions/roles feels like a meaningful step forward. The contribution might be more primarily one of method/analysis, not a new ecological law.

SUPPORT OF CLAIMS/CONCLUSIONS

- Some comparative claims might make use of additional statistics. Phrases such as "contrasting patterns" between pollutant groups should be backed by named tests, effect sizes, and multiple-testing corrections.
- Please provide explicit quantitative definitions for degrader/helper/potentiator, especially potentiator. I recommend introducing and reporting a potentiator index metric/score of some kind across conditions that will facilitate comparison/ranking. Perhaps: $[Performance(DH+P) - Performance(DH)] / Performance(DH)$. Where P = Potentiator and DH = Degrader+Helper.
- "Ablation" test: For the best trio, experimentally validate DH vs DH+P and swap P for a near-neighbor to demonstrate role specificity rather than a generic third-member effect.
- In silico sensitivity test: Provide a brief GSMM sensitivity analysis (e.g., $\pm 20\%$ on exchange costs or carbon availability) to show robustness of model-driven recommendations.
- Some bounding of the generalization claims is probably appropriate. If results are shown in one medium/regime, either (i) replicate a key result under one orthogonal condition or (ii) narrow the language to the tested context.

METHODOLOGY

- Generally sound experimental and modeling choices.
- Units/quantification: Standardize growth/flux units and annotate axes; state detection limits explicitly. Replace "complete removal" with "< LOD (method, value)".
- Reporting: Ensure all figure panels referenced in text (e.g., 1a/1b/3c) are explicitly called out.

ORGANIZATION AND CLARITY

- Bridge early analyses to the six focal isolates. Add a short paragraph/schematic explicitly linking Figs 1–2 to species selection (the focus of the rest of the paper).
- Figure hygiene: The paper felt a bit like a stream of consciousness. I'd start with a "Results map" table linking each scientific question to the figure/panel that answers it and drive the paper through more of a narrative style. It reads more like a huge dump of a tremendous amount of (good) methodology and analysis.

BOTTOM LINE

Considerable revisions needed. Of particular note: (1) operational role definitions with a reported potentiator index of some kind (2) improved bridging/organization, and (3) the small ablation and in-silico sensitivity checks. If these are addressed thoroughly in text/analysis (and the vitamin mechanism is framed cautiously), I would be comfortable moving to acceptance.

(Remarks on code availability)

Version 1:

Reviewer comments:

Reviewer #1

(Remarks to the Author)

The authors have satisfactorily addressed all of my previous comments. I find that the revisions have improved the clarity and quality of the manuscript, and I have no further concerns at this stage.

(Remarks on code availability)

Reviewer #2

(Remarks to the Author)

The authors have addressed all my concerns.

(Remarks on code availability)

Reviewer #3

(Remarks to the Author)

The revision is improved and sound. The main contribution is methodological—using multistrain GSMM/FBA to guide role-based community design—with experimental results that broadly align with model predictions. The work is significant as an analysis/toolkit advance for engineering multi-member consortia under mixed-pollutant conditions. I support publication.

(Remarks on code availability)

Potentiator index: State the definition explicitly as the DH vs. DH+P contrast (or justify any difference), pre-specify one primary performance metric (growth AUC or % removal at fixed time), and report CIs.

Statistics: Surface the multiple-testing approach (e.g., FDR) in main text/captions and pair comparative statements with named tests and effect sizes/CIs.

Scope: Bound generalization claims to the tested medium/regime (or point clearly to any added orthogonal replication).

Mechanism wording: Keep vitamin/cofactor effects as “consistent with” unless a minimal auxotrophy/complementation or uptake-inhibition test is included.

Sensitivity: Add a sentence confirming whether rankings/index values remain stable under the reported parameter perturbations.

With these small edits, I recommend acceptance.

Reviewers' comments:

Reviewer #1 (Remarks to the Author):

GENERAL COMMENT

This manuscript by Ruan et al. presents an ambitious and comprehensive study integrating experimental microbiome acclimation, multi-omics analyses, and genome-scale metabolic modeling (GSMM) via the SuperCC framework to rationally design synthetic microbiomes optimized for complex pollutant remediation. The authors propose the novel “Degradere–Helper–Potentiator” (DHP-Com) paradigm, validate it with multi-strain culture experiments, untargeted metabolomics, and multi-pollutant degradation assays, and link their findings to mechanistic modeling predictions.

Noteworthy results: The work identifies keystone “potentiator” strains that enhance pollutant removal and community resilience across multiple contaminant classes, demonstrates that co-contamination can foster greater microbial diversity and metabolic cross-feeding, and introduces SuperCC as a modeling framework capable of simulating realistic community dynamics under multi-pollutant conditions.

Significance: The study addresses an important gap in synthetic microbiome design for real-world co-contamination scenarios, rather than idealized single-pollutant setups, and is likely to be of interest to researchers in environmental biotechnology, microbial ecology, and systems/synthetic biology.

Answer: We sincerely thank Reviewer 1 for the positive and encouraging comments on our work. We truly appreciate your recognition of the study’s significance and contributions, and your thoughtful feedback motivates us to further refine and strengthen the manuscript.

Relation to established literature: The DHP-Com paradigm extends prior “Degradere–Helper” concepts (e.g., refs. 13–14, 15) by explicitly incorporating potentiator functions identified via integrated modeling and metabolomics. The work builds on, but clearly advances, existing studies such as Smith et al. 2024 and Ruan et al. 2024.

Support for conclusions: Overall, the experimental results and modeling outputs support the major conclusions. However, some claims, in particular, regarding potentiator roles, are inferred rather than directly demonstrated and would benefit from additional validation or caveats (see Major Comments 8–9).

Answer: We sincerely appreciate Reviewer 1’s thoughtful and constructive comments, as well as the recognition of how our study extends the previous “Degradere–Helper” concept. The roles of potentiator strains are indeed supported by multiple lines of evidence, including model predictions, metabolomics analyses, and experiments with potentiator-deletion communities. We have elaborated on these points in detail in our responses to Major Comments 8–9.

Methodological soundness: The experimental design is diverse and generally robust, but certain aspects of the modeling approach (e.g., GEM reconstruction choices, simulation assumptions, parameter sensitivity) require clearer justification. The core methodology is sound for the field, but improved transparency would strengthen reproducibility and trust in the results.

Answer: We sincerely thank Reviewer 1 for the thoughtful comments. While we appreciate the recognition of our overall methodological soundness, we fully agree that clearer justification of modeling choices and assumptions is critical for transparency and reproducibility. We have thus

revised the manuscript to provide additional explanations, with detailed elaboration on these points in our responses to Major Comments 2, 3, and 10.

Detail for reproducibility: The Methods section is extensive, but some parameters (exchange bounds, biomass coefficients, gap-filling criteria, metabolite identification thresholds) should be explicitly stated to fully meet reproducibility standards.

Answer: We thank Reviewer 1 for this valuable comment. We agree that providing explicit parameter details is essential for reproducibility. In our analysis, significantly different metabolites were first identified using fold change ≥ 1 , $p < 0.05$, and $VIP \geq 1$ (OPLS-DA), as shown in Fig. 4B. To further highlight the most critical metabolites, we applied more stringent criteria ($p < 0.01$, $VIP \geq 2$), with results presented in Fig. 4C, S4, and S5. These thresholds have now been clearly stated in the Methods section and figure legends. For the metabolic model, we performed complete and detailed modification in accordance with the reference (47. Thiele, I., & Palsson, B. O. (2010). A protocol for generating a high-quality genome-scale metabolic reconstruction. *Nature Protocols*, 5(1), 93–121.) to ensure that the simulation results are consistent with those obtained in the experiment. As suggested, we have revised the Methods section to clearly describe these thresholds, along with additional parameter details.

On Lines 452-465, we write:

“...Only reactions supported by genetic evidence in the genome of the target strain or its closely related species were retained... By reviewing the literature, we modified the direction of reversible reactions and the mass-charge balance of metabolic reactions, and modified reactions utilizing CO₂ and reversible CO₂ reactions into unidirectional reactions... Except for the essential nitrogen source (NH₄⁺) which was set to 100 mmol/gDW, the contents of all other inorganic substances without carbon (C) or nitrogen (N) were set to 1000 mmol/gDW.”

MAJOR COMMENTS

1. Model Transparency and Assumptions

Lines 431–472

How were the uptake bounds (LB) and secretion bounds (UB) for each metabolite determined for single-strain and community models? Were they uniform across all strains or strain-specific, based on annotation? Please clarify how uptake (LB) and secretion (UB) bounds were determined.

Answer: Thanks for the reviewer’s comments. The determination of uptake bounds (LB) and secretion bounds (UB) for each metabolite was based on the protocol described in the reference (47. Thiele, I., & Palsson, B. O. (2010). A protocol for generating a high-quality genome-scale metabolic reconstruction. *Nature Protocols*, 5(1), 93–121.)

For single-strain models:

First, the medium was configured according to the exchange metabolites included in each single-strain metabolic model. Among the components, the minimal carbon source (e.g., glucose) and NH₄⁺ were set with an LB of 100 mmol/gDW; all other organic substances were set with an LB of 0 mmol/gDW; and all other inorganic substances without carbon (C) or nitrogen (N) were set to the maximum value (an LB of 1000 mmol/gDW). The UB for all substances was uniformly set to 1000 mmol/gDW. Through this configuration, the minimal medium for each single strain was determined. Notably, in the single-strain medium, the LB of CO₂ was set to 0 mmol/gDW, and the

UB of O₂ was set to 0 mmol/gDW.

For community models:

The medium was the union of the minimal media of all individual strains in the community. Similar to the single-strain setup, only the target carbon sources were specifically configured (e.g., configurations for C = 1 and C = 4 conditions), while NH₄⁺ still had an LB of 100 mmol/gDW, all other organic substances had an LB of 0 mmol/gDW, and all other inorganic substances without C/N had an LB of 1000 mmol/gDW. The UB for all substances was consistently set to 1000 mmol/gDW, thereby determining the minimal medium for the microbial community. Additionally, the LB of CO₂ (0 mmol/gDW) and UB of O₂ (0 mmol/gDW) were maintained the same as in the single-strain medium.

The above detailed information has been supplemented in the Supplementary Text.

Did you use any thermodynamic feasibility filtering (e.g., TFA, min–max driving force) to remove energetically infeasible flux cycles? If not, discuss whether including TFA would alter predictions of cross-feeding patterns.

Answer: Thanks for the reviewer’s comments. In our previous study (Ruan, Z. et al. Engineering natural microbiomes toward enhanced bioremediation by microbiome modeling. *Nat Commun* 15, 4694 (2024).), we conducted thermodynamic validation for several reactions involved in pollutant metabolism and confirmed that both our modeling workflow and the SuperCC framework simulations comply with thermodynamic laws. Thus, we consider this modification process to be relatively scientific. Furthermore, the draft single-strain models we constructed were developed using ModelSEED—a database built by integrating chemical data from multiple sources, applying standardized transformations, identifying redundancies, and calculating thermodynamic properties. Thus, thermodynamic feasibility filtering was already incorporated during the initial model construction phase. Additionally, when manually refining these draft models, the metabolic reaction databases we utilized were also sourced from ModelSEED, which ensured that every reaction we added complies with thermodynamic principles.

Reference: Seaver, S. M. D. et al. The ModelSEED Biochemistry Database for the integration of metabolic annotations and the reconstruction, comparison and analysis of metabolic models for plants, fungi and microbes. *Nucleic Acids Res* 49, D575–D588 (2021).

For exchange metabolite inclusion, what criteria were used (e.g., detected in metabolomics, KEGG annotation, gap-filling output)? Were dead-end metabolites removed from the community network?

Answer: Thanks for the reviewer’s comments. Exchange metabolites were selected based on evidence from metabolomics identification, single-strain modeling, and multi-strain modeling. To improve clarity, we have added Supplementary Table S6, which details the criteria for inclusion of exchange metabolites.

On Table S6, we write:

“Supplementary Table S6 Identification of key metabolites required for experimental validation via multiple analytical methods. ER, exchange reactions. Scenario 1: Each strain attains equal biomass growth; Scenario 2: Each strain within the community achieves growth (community-level optimal solution). For the exchange metabolites in multi-strain modeling, opposite patterns (OP) were observed: one was uptake (when C = 4), and the other was release (when C = 1). Using the

SuperCC modeling framework, the exchange metabolites were simulated for six-strain combinations under two nutritional scenarios: (1) Minimal Medium (MM) supplemented with 100 mmol/gDW glucose (single carbon source, corresponding to C = 1); and (2) MM with 25 mmol/gDW each of glucose, citrate, acetate, and fumarate (quadruple carbon sources, corresponding to C = 4). Each metabolite was experimentally validated by supplementing it to the microbial consortium composed of strains A1, P1, C2, S4, B4, and L3, which was inoculated into MM-10% P medium containing 10 ppm of TC and OTC. After 2 days of incubation, the residual concentrations of TC and OTC were measured, and the enhancement of degradation was calculated relative to the control without metabolite supplementation. TC, tetracycline; OTC, oxytetracycline. The data are presented as mean values (n = 4 biological independent replicates).

Category	Compounds	Identified by metabolomics	Single-strain modeling	Multi-strain modeling	Enhanced degradation on TC	Enhanced degradation on OTC
Amino Acids and Derivatives	L-Proline	√	√ (ER occur only in strains A1, P1, C2, and L3)	/	+3.98%	+4.34%
Amino Acids and Derivatives	N-Formyl-L-methionine	√	/	/	+1.50%	-0.24%
Amino Acids and Derivatives	L-Glutamate	/	√ (ER occur in all six strains)	√ (OP appear in Scenario 1; ER occur in Scenario 1 and 2)	+8.98%	+6.86%
Amino Acids and Derivatives	L-Phenylalanine	√	√ (ER occur only in strains A1, P1, C2, and S4)	√ (OP appear in Scenario 1; ER occur in Scenario 1 and 2)	+5.78%	+3.61%
Amino Acids and Derivatives	Glycine	/	√ (ER occur only in strains A1, P1, C2, and S4)	√ (OP appear in Scenario 2; ER occur in Scenario 1 and 2)	+8.98%	+8.27%

Amino Acids and Derivatives	L-Histidine	/	√ (ER occur only in strain A1)	√ (OP appear in Scenario 1 and 2)	+2.71%	+0.85%
Vitamins	Vitamin B3 (niacin)	√	√ (ER occur only in strain A1)	/	+21.36%	+6.43%
Vitamins	Vitamin B1	√	√ (ER occur only in strains A1, P1, C2, and L3)	√ (OP appear in Scenario 2; ER only occur in Scenario 2)	+18.91%	+7.70%
Vitamins	Vitamin D2	√	/	/	+15.66%	+8.69%
Organic Acids	Palmitate	√	√ (ER occur only in strains A1, P1, B4, and S4)	√ (OP appear in Scenario 2; ER only occur in Scenario 2)	+5.91%	+4.06%
Organic Acids	Myristic acid	/	√ (ER occur only in strains A1, P1, L3, B4, and S4)	√ (OP appear in Scenario 2; ER only occur in Scenario 2)	+12.85%	+8.02%
Organic Acids	Succinate	√	√ (ER occur only in strains A1, P1, C2, and S4)	/	+15.22%	+9.49%

How were biomass functions standardized and strain-specific precursors handled?

Answer: Thanks for the reviewer’s comments. The standardization of bacterial biomass function was established based on the ModelSEED database, using the protein sequences derived from the uploaded bacterial genome. This information has been included in Supplementary Table S4.

Regarding strain-specific precursors, those that can be acquired via exchange reactions—such as inorganic compounds without carbon or nitrogen—were directly supplied from the defined culture medium in the model. For other required precursors, the model simulated their synthesis through the internal metabolic network of the bacterium, using a unified nitrogen source (NH₄⁺) and different carbon sources (with simulations performed for both C = 1 and C = 4 compounds). This allowed the bacterium to generate the necessary components for biomass production.

On Table S4, we write:

“Supplementary Table S4 Biomass reaction composition for the reconstructed metabolic models. For each species, the upper section lists the ModelSEED compound identifiers for each component,

2. Community Model Configuration

Lines 453–472

Were biomass weighting coefficients (c_k in Eq. [4]) equal for all species, or did you use empirically measured relative abundances to weight objectives?

Answer: Thanks for the reviewer's comments. Four commonly used scenarios for setting the biomass weighting coefficients c_k in Eq. [4] have been added to the Methods section. The biomass weighting coefficients were not necessarily equal for all species; empirically measured relative abundances could also be utilized as weights via the "defined abundances" scenario.

On Lines 496-500, we write:

“Four commonly used scenarios were provided, including 1) equal abundance for each organism; 2) no limitation for any organisms, meaning that the biomass of each organism is allowed to be zero; 3) defining biomass of a specific organism as community biomass (used for identifying organisms in a community that could improve the growth of the target organism); and 4) any defined abundances.”

3. Simulation Scenarios and Parameter Sensitivity

Lines 191–215

In the C=1 vs C=4 scenarios, did you hold total carbon molar input constant or increase total available carbon? If the latter, could increased total resource availability (not diversity) drive biomass gains?

Answer: Thanks for the reviewer's comments. We appreciate the reviewer for raising this critical point, as clarifying carbon input conditions is essential to validating the effect of carbon diversity. Below we provide explicit carbon-mole calculations, along with the rationale for our experimental design:

1) Carbon input calculation:

A. Single-carbon (C = 1) scenario:

- 100 mmol glucose ($C_6H_{12}O_6$, containing 6 carbon atoms per mole) \rightarrow Total carbon input = 100 mmol \times 6 = 600 mmol C

B. Four-carbon (C = 4) scenario:

We selected four representative carbon sources and set their individual concentrations to 25 mmol (to ensure balanced representation of each carbon type). The total carbon input is calculated as follows:

- 25 mmol glucose (6 C atoms) $\rightarrow 25 \times 6 = 150$ mmol C
 - 25 mmol citrate (6 C atoms) $\rightarrow 25 \times 6 = 150$ mmol C
 - 25 mmol acetate (2 C atoms) $\rightarrow 25 \times 2 = 50$ mmol C
 - 25 mmol fumarate (4 C atoms) $\rightarrow 25 \times 4 = 100$ mmol C
- Total carbon input (for C = 4) = 150 + 150 + 50 + 100 = 450 mmol C

2) Rationale for carbon input design

We intentionally set the total carbon molar input of the C = 4 scenario (450 mmol C) lower than that of the C = 1 scenario (600 mmol C). This conservative design was specifically intended to eliminate the confounding effect of "increased total carbon availability"—if biomass still increased in the C = 4 condition despite lower total carbon, it would more strongly confirm that the observed biomass gains originate from carbon diversity/quality, metabolic routing, or interspecific community

interactions, rather than resource quantity.

3) Supplementary updates in the manuscript

To ensure transparency, we have included a summary text (Supplementary Text) that clearly lists the molar concentration, carbon atom number per molecule, and individual total carbon contribution of each carbon source in both C = 1 and C = 4 scenarios. Readers can directly refer to these sections to verify the carbon input difference and our experimental design logic.

On Supplementary Text (Modeling Media Part), we write:

“For community models:

The medium was the union of the minimal media of all individual strains in the community. Similar to the single-strain setup, only the target carbon sources were specifically configured (e.g., configurations for C = 1 and C = 4 conditions), with details as follows:

C = 1 condition: 100 mmol glucose (C₆H₁₂O₆, containing 6 carbon atoms per mole) was used as the sole carbon source, resulting in a total carbon input of 100 mmol × 6 = 600 mmol C.

C = 4 condition: Four carbon sources were included, each at 25 mmol to ensure balanced representation: 25 mmol glucose (6 C atoms, contributing 150 mmol C), 25 mmol citrate (6 C atoms, contributing 150 mmol C), 25 mmol acetate (2 C atoms, contributing 50 mmol C), and 25 mmol fumarate (4 C atoms, contributing 100 mmol C). The total carbon input for the C = 4 condition was 150 + 150 + 50 + 100 = 450 mmol C.

While NH₄⁺ still had an LB of 100 mmol/gDW, all other organic substances had an LB of 0 mmol/gDW, and all other inorganic substances without C/N had an LB of 1000 mmol/gDW. The UB for all substances was consistently set to 1000 mmol/gDW, thereby determining the minimal medium for the microbial community. Additionally, the LB of CO₂ (0 mmol/gDW) and UB of O₂ (0 mmol/gDW) were maintained the same as in the single-strain medium.”

Did you perform sensitivity analysis on carbon source composition to see if results were robust to the specific four-carbon mix used (glucose, citrate, acetate, fumarate)?

Answer: We sincerely thank Reviewer 1 for this valuable suggestion. In response, we performed a sensitivity analysis on the carbon source composition to evaluate the robustness of our results with respect to the specific four-carbon mix (glucose, citrate, acetate, fumarate). Specifically, we used a baseline medium containing 20 mM glucose, 20 mM acetate, 20 mM citrate, and 20 mM fumarate, and inoculated it with the microbial consortium (strains A1, P1, C2, L3, S4, B4). Based on this baseline, we designed two types of perturbation experiments:

Single-factor perturbation: To assess the effect of fluctuations in individual carbon sources, we adjusted the concentration of one carbon source by ±10% (i.e., increased or decreased by 10%) while keeping the others constant.

We calculated the relative biomass change as:

$$\text{Relative change rate} = (\text{Biomass_perturbed} - \text{Biomass_baseline}) / \text{Biomass_baseline}$$

and defined the robustness coefficient as $R = 1 - |\text{Relative change rate}|$ (where the absolute value accounts for both increases and decreases in biomass).

Multi-factor perturbation: To assess robustness under combined fluctuations, we simultaneously perturbed the concentrations of all four carbon sources by ±10% (consistent with the amplitude used in single-factor tests).

In both scenarios, the system was considered robust if the relative change rate remained <10% and

the robustness coefficient (R) was close to 1. Our results (summarized in Supplementary Table S8) showed that under all perturbation conditions, the relative biomass change rate ranged from -8.67% to 8.58% (all <10%), and R values were between 0.9133 and 0.9985 (close to 1).

In the model, we also conducted a robustness test: we set the carbon source content to fluctuate by $\pm 20\%$, and used SuperCC to simulate and predict the optimal biomass of the six-bacterium community model. Our results (summarized in Supplementary Table S9) showed that under all perturbation conditions, the relative biomass change rate ranged from -0.03% to 0.07% (all < 1%), and R values were close to 1. These findings confirm that our results are robust to variations in the specific four-carbon mix used.

On Supplementary Table S8, we write:

“Supplementary Table S8 Experimental validation of sensitivity analysis for carbon source composition. Bacterial growth was monitored for 1800 min.

Condition (Carbon sources) mM				Mean Biomass	SD	Relative Change rate (%)	Robustness Coefficient (R)
Glucose	Acetate	Citrate	Fumarate				
20	20	20	20	0.7301	0.0160	–	–
22	20	20	20	0.7094	0.0243	-2.84%	0.9716
18	20	20	20	0.7326	0.0224	+0.34%	0.9966
20	22	20	20	0.7280	0.0315	-0.29%	0.9971
20	18	20	20	0.7928	0.0335	+8.58%	0.9142
20	20	22	20	0.6668	0.0523	-8.67%	0.9133
20	20	18	20	0.6942	0.0424	-4.92%	0.9508
20	20	20	22	0.7752	0.0441	+6.17%	0.9383
20	20	20	18	0.7078	0.0172	-3.06%	0.9694
22	22	18	18	0.7170	0.0232	-1.80%	0.982
18	18	22	22	0.7266	0.0246	-0.48%	0.9952
22	18	18	22	0.7030	0.0196	-3.72%	0.9628
18	22	22	18	0.7312	0.0181	+0.15%	0.9985

Note: Relative change rate = $(\text{Biomass}_{\text{perturbed}} - \text{Biomass}_{\text{baseline}}) / \text{Biomass}_{\text{baseline}} \times 100\%$

and defined the robustness coefficient as $R = 1 - |\text{Relative change rate}|$ (where the absolute value accounts for both increases and decreases in biomass). ”

On Supplementary Table S9, we write:

“Supplementary Table S9 Modeling validation of sensitivity analysis of DHP-Com consortium for carbon source composition.

Condition (Carbon sources)mmol/g DW				Optimal Biomass	Relative Change rate (%)	Robustness Coefficient (R)
Glucose	Citrate	Acetate	Fumarate			
25	25	25	25	421.812654823115	-	-
20	25	25	25	421.672057879753	-0.033331609%	0.999666684
30	25	25	25	421.953251766474	0.033331609%	0.999666684
25	20	25	25	421.812654823114	-2.42568E-13%	1
25	30	25	25	421.765789175328	-0.011110536%	0.999888895
25	25	20	25	421.765789175328	-0.011110536%	0.999888895
25	25	30	25	421.859520470903	0.011110536%	0.999888895
25	25	25	20	421.718923527538	-0.022221072%	0.999777789
25	25	25	30	421.906386118687	0.022221072%	0.999777789
20	20	30	30	421.812654823113	-4.7166E-13%	1
30	30	20	20	421.812654823071	-1.04304E-11%	1
20	30	20	30	421.718923527539	-0.022221072%	0.999777789
30	20	30	20	421.906386118687	0.022221072%	0.999777789
20	30	30	20	421.625192231966	-0.04442145%	0.999555579
30	20	20	30	422.000117414260	0.04442145%	0.999555579
20	20	20	20	421.531460936395	-0.066663217%	0.999333368
30	30	30	30	422.093848709832	0.066663217%	0.999333368

Note: Except for the change in the content of carbon source C, the contents of other inorganic substances remain unchanged. In particular, the content of NH_4^+ is always maintained at 100 mmol/g DW. ”

For the network visualizations in Fig. 5c, how was the “number of exchanged compounds” thresholded? Were only flux-carrying exchanges included, or all feasible exchanges under optimal growth?

Answer: Thanks for the reviewer’s comments. Regarding the network visualizations in Fig. 5c, the "number of exchanged compounds" was determined by including all feasible exchanges under simulated optimal growth conditions. Specifically, an exchanged compound was counted and visualized only if it was secreted by one species and subsequently taken up by at least one other species in the community. Compounds that were secreted but not consumed by any organism were excluded from the analysis and the final figure.

4. Validation of Model Predictions

Lines 104–117

Were biomass predictions validated against experimental growth rate measurements for any subset of strains? If not, can you provide at least qualitative agreement with rank-order abundances?

Answer: Answer: Thank you for your insightful comment. We conducted correlation analyses between the abundances of the Top 1-10 and Top 1-30 species across the three treatments and their maximum biomass under optimal environmental conditions. The results revealed linear correlations in all cases. Notably, except for the TC group's Top 1-10 (where no correlation was observed), the other groups showed a decreasing slope in the linear relationship as lower-abundance species (Top 20-30) were included compared to the Top 1-10. This indicates that species with lower abundance rankings exhibited greater optimal biomass at the single-strain level, corresponding to relatively slower growth rates. These findings provide qualitative agreement between biomass predictions and rank-order abundances.

On Fig. S3, we write:

“

Fig. S3 | Correlation analysis between the abundances of Top 1-10 and Top 1-30 species and their maximum biomass under optimal conditions (simulated by FBA) across three treatments: TC, OTC, and TC&OTC. TC, tetracycline; OTC, oxytetracycline. CK, control group. The data of the abundances are presented as mean values.”

For the observed “fast-growing, low-biomass strains” in co-contamination (line 116), could this be an artifact of model biomass equation stoichiometry?

Answer: We sincerely thank the reviewer for this insightful question. We agree that the observed "fast-growing, low-biomass" trend could, in principle, be influenced by the stoichiometric formulation of the model biomass equation. To address this concern, we conducted experimental validation using the top 50 strains identified under the TCs co-contamination condition. Specifically, we grouped these strains and selected representative isolates from each cluster for further testing. The experiments confirmed that strains with a steeper slope in the early growth phase (i.e., faster growth rates) entered the stationary phase earlier and reached a lower final biomass. As shown in Figure S9, these results are consistent with the model predictions.

On Fig. S9, we write:

“

Fig. S9 | Growth curves of representative species from the co-contamination of the top 50 keystone strains. Bacterial growth curves were determined in an inorganic salt medium supplemented with 1 mol/L glucose. The strains were divided into three subsets based on their predicted ranking: a, growth curves of three representative strains from the Top 1-10 subset; b, growth curves of three representative strains from the Top 11-30 subset; c, growth curves of three representative strains from the Top 31-50 subset. The data are presented as mean values ($n = 4$ biological independent replicates).”

5. Flux Predictions and Interpretation

Lines 205–215

Did you differentiate between obligatory vs facultative exchanges in the flux predictions? For example, were some metabolites exchanged only under $C=1$ but not under $C=4$?

Answer: Thanks for the reviewer’s comments. We did not specifically differentiate between obligatory and facultative exchanges in our flux prediction analyses. Microbial metabolic exchanges are inherently dynamic, as they are influenced by redundant genes within the community and environmental conditions—for instance, variations in carbon/nitrogen sources or the composition of microbial consortia can all induce changes in exchanged metabolites. In the context of this study, the difference in the carbon source type of the medium directly drove distinct metabolic exchange reactions among strains.

Notably, while metabolic exchanges varied between the $C = 1$ and $C = 4$ conditions (both aimed at achieving optimal overall microbial community biomass), a certain degree of commonality in exchanged metabolites was observed: the proportions of shared exchanged metabolite categories were 86.11% under $C = 1$ and 93.94% under $C = 4$, respectively. We also found that $C = 1$ harbored a greater number of total exchanged metabolites and unique exchanged metabolites compared to $C = 4$. Specifically, metabolites including cytosine, fumarate, glycerol-3-phosphate, acetate, and palmitate were detected as exchanged metabolites under $C = 1$ but absent under $C = 4$; conversely,

urea and nicotinamide ribonucleotide were only identified as exchanged metabolites under C = 4 and not under C = 1.

All these condition-specific differences in exchanged metabolites have been annotated in the source data.

The identification of B4 as an “altruistic keystone” (line 213) is compelling — did you test whether this classification holds under different objective functions (e.g., community-wide substrate uptake minimization)?

Answer: Thanks for the reviewer’s comments. In our model, we conducted simulations under different substrate conditions and evaluated multiple objective functions—including scenarios where (1) each strain within the community achieves growth (community-level optimal solution), and (2) each strain attains equal biomass growth. Across all these objective functions, B4 consistently maintained its classification as an "altruistic keystone strain": it secreted more metabolites to promote the growth of other strains in the community. The results of these additional analyses have been supplemented in Table S5.

Furthermore, our classification of B4 as an altruistic keystone strain is not solely based on predictions from metabolic modeling. Metabolomic experiments also supported this conclusion: under single pollutant exposure and two pollutant exposure scenarios, both B4 and S4 exhibited metabolic redundancy. This redundancy resulted in fewer unique metabolites being produced when these altruistic keystone strains were present in the community. To reinforce this finding, we have supplemented metabolomic data of the microbial community under single-pollutant conditions in Fig. S6.

On Fig. S6, we write:

“

Fig. S6 | Venn diagram and functional annotation of metabolites. *a*, The differences in microbial metabolites of TCs-group with or without strain B4/S4 in the synthetic microbiomes. *b*, The differences in microbial metabolites of TCs-group with or without strain A1/P1/C2 in the synthetic

microbiomes. c, The differences in microbial metabolites of OTC-group with or without strain B4/S4 in the synthetic microbiomes. d, The differences in microbial metabolites of OTC-group with or without strain A1/P1/C2 in the synthetic microbiomes. Circular sectors display HMDB-annotated metabolite classes. TCs, tetracycline&oxytetracycline (vs CK; $p < 0.05$, $VIP \geq 1$).”

On Table S5, we write:

“**Supplementary Table S5** Flux balance analysis illustrating interspecies metabolic exchange under the objective of equalizing biomass for each species in the community. Using the SuperCC modeling framework, the maximum biomass was simulated for six-strain combinations under two nutritional scenarios: (1) MM supplemented with 100 mmol/gDW glucose (single carbon source, $C = 1$); and (2) MM with 25 mmol/gDW each of glucose, citrate, acetate, and fumarate (quadruple carbon sources, $C = 4$). Substances with a light blue background represent metabolites unique to the $C = 4$ condition.

C=1					
Secretion	Compounds	Formula	ModelSEED ID	Absorption	Category
A1	L-Lysine	C6H15N2O2	cpd00039	P1, C2, L3	Amino Acids and Derivatives
	Octadecanoic acid	C18H35O2	cpd01080	P1, C2, S4	Organic Acids
	(R)-3-Hydroxybutanoate	C4H7O3	cpd00797	C2	Organic Acids
	D-Glucosamine	C6H14NO5	cpd00276	P1	Saccharides and Derivatives
	Succinate	C4H4O4	cpd00036	C2, S4	Organic Acids
	L-Glutamate	C5H8NO4	cpd00023	C2, L3, S4	Amino Acids and Derivatives
L3	Xanthine	C5H4N4O2	cpd00309	P1, C2	Nucleosides/Nucleotides and Derivatives
	D-Alanine	C3H7NO2	cpd00117	A1, C2	Amino Acids and Derivatives
	Glycine	C2H5NO2	cpd00033	A1, P1	Amino Acids and Derivatives
	Urea	CH4N2O	cpd00073	C2	Organic Acids
	Ornithine	C5H13N2O2	cpd00064	P1	Nucleosides/Nucleotides and Derivatives
	L-Proline	C5H9NO2	cpd00129	A1, P1	Amino Acids and Derivatives
	L-Serine	C3H7NO3	cpd00054	A1, C2	Amino Acids and Derivatives
B4	Glycerol	C3H8O3	cpd00100	P1	Saccharides and Derivatives
	Acetate	C2H3O2	cpd00029	A1, P1, S4	Organic Acids
	Octadecanoic acid	C18H35O2	cpd01080	P1, C2, S4	Organic Acids
	L-Glutamate	C5H8NO4	cpd00023	C2, L3, S4	Amino Acids and Derivatives
	Uridine	C9H12N2O6	cpd00249	P1	Nucleosides/Nucleotides and Derivatives
	Inosine	C10H12N4O5	cpd00246	P1	Nucleosides/Nucleotides and Derivatives
	Acetate	C2H3O2	cpd00029	A1, P1, S4	Organic Acids
P1	2-Oxoglutarate	C5H4O5	cpd00024	P1, C2	Organic Acids
	L-Lysine	C6H15N2O2	cpd00039	P1, C2, L3	Amino Acids and Derivatives
	L-Tyrosine	C9H11NO3	cpd00069	C2	Amino Acids and Derivatives
	L-Phenylalanine	C9H11NO2	cpd00066	A1, C2	Amino Acids and Derivatives
	Uracil	C4H4N2O2	cpd00092	A1, C2, L3	Nucleosides/Nucleotides and Derivatives

	Myristic acid	C14H27O2	epd03847	L3, B4, S4	Organic Acids
	Hypoxanthine	C5H4N4O	epd00226	A1, L3, S4	Nucleosides/Nucleotides and Derivatives
	N-Acetyl-D-glucosamine	C8H15NO6	epd00122	S4	Saccharides and Derivatives
	Cytidine	C9H13N3O5	epd00367	B4	Nucleosides/Nucleotides and Derivatives
	L-Arginine	C6H15N4O2	epd00051	L3	Amino Acids and Derivatives
	D-Alanine	C3H7NO2	epd00117	A1, C2	Amino Acids and Derivatives
	L-Serine	C3H7NO3	epd00054	A1, C2	Amino Acids and Derivatives
	D-Trehalose	C12H22O11	epd00794	S4	Saccharides and Derivatives
	Adenosine	C10H13N5O4	epd00182	B4	Nucleosides/Nucleotides and Derivatives
	L-Glutamate	C5H8NO4	epd00023	C2, L3, S4	Amino Acids and Derivatives
	Succinate	C4H4O4	epd00036	C2, S4	Organic Acids
	D-Mannose	C6H12O6	epd00138	A1, C2	Saccharides and Derivatives
C2	Thiamin	C12H17N4OS	epd00305	A1, P1, L3	Vitamins
	L-Tryptophan	C11H12N2O2	epd00065	P1, B4	Amino Acids and Derivatives
	L-Proline	C5H9NO2	epd00129	A1, P1	Amino Acids and Derivatives
	Fumarate	C4H2O4	epd00106	A1, P1	Saccharides and Derivatives
	Acetate	C2H3O2	epd00029	A1, P1, S4	Organic Acids
	D-Fructose	C6H12O6	epd00082	A1, P1	Saccharides and Derivatives
	Glycine	C2H5NO2	epd00033	A1, P1	Amino Acids and Derivatives
	D-Glucosamine	C6H14NO5	epd00276	P1	Saccharides and Derivatives
S4	Hypoxanthine	C5H4N4O	epd00226	A1, L3, S4	Nucleosides/Nucleotides and Derivatives
	Nicotinamide ribonucleotide	C11H14N2O8P	epd00355	P1	Vitamins
	Palmitate	C16H31O2	epd00214	A1	Organic Acids
	(R)-3-Hydroxybutanoate	C4H7O3	epd00797	C2	Organic Acids
	D-Glucosamine	C6H14NO5	epd00276	P1	Saccharides and Derivatives
	D-Mannose	C6H12O6	epd00138	A1, C2	Saccharides and Derivatives
	Fumarate	C4H2O4	epd00106	A1, P1	Saccharides and Derivatives

C=4					
Secretion	Compounds	Formula	ModelSEED ID	Absorption	Category
A1	Octadecanoic acid	C18H35O2	epd01080	P1, B4, S4	Organic Acids
	Succinate	C4H4O4	epd00036	C2, S4	Organic Acids
	Hypoxanthine	C5H4N4O	epd00226	C2, L3, S4	Nucleosides/Nucleotides and Derivatives
	L-Lysine	C6H15N2O2	epd00039	P1, C2, L3	Amino Acids and Derivatives
B4	D-Glucosamine	C6H14NO5	epd00276	P1, S4	Saccharides and Derivatives
	Inosine	C10H12N4O5	epd00246	P1	Nucleosides/Nucleotides and Derivatives
	Uridine	C9H12N2O6	epd00249	P1	Nucleosides/Nucleotides and Derivatives
	Deoxyuridine	C9H12N2O5	epd00412	P1	Nucleosides/Nucleotides and Derivatives
	L-Lysine	C6H15N2O2	epd00039	P1, C2, L3	Amino Acids and Derivatives
C2	2-Oxoglutarate	C5H4O5	epd00024	P1	Organic Acids
	Thiamin	C12H17N4OS	epd00305	A1, P1, L3	Vitamins
	4-Aminobutanoic acid	C4H9NO2	epd00281	L3	Amino Acids and Derivatives
	L-Tryptophan	C11H12N2O2	epd00065	P1, B4	Amino Acids and Derivatives
	Uracil	C4H4N2O2	epd00092	A1, L3	Nucleosides/Nucleotides and Derivatives
	Xanthinethine	C5H4N4O2	epd00309	A1, P1	Nucleosides/Nucleotides and Derivatives
	(R)-3-Hydroxybutanoate	C4H7O3	epd00797	A1	Organic Acids
	L-Proline	C5H9NO2	epd00129	A1, P1	Amino Acids and Derivatives
	Glycine	C2H5NO2	epd00033	A1, P1	Amino Acids and Derivatives
	D-Glucosamine	C6H14NO5	epd00276	P1, S4	Saccharides and Derivatives
L3	Cytosine	C4H5N3O	epd00307	C2	Nucleosides/Nucleotides and Derivatives
	L-Proline	C5H9NO2	epd00129	A1, P1	Amino Acids and Derivatives
	Glycine	C2H5NO2	epd00033	A1, P1	Amino Acids and Derivatives
	D-Alanine	C3H7NO2	epd00117	A1, C2	Amino Acids and Derivatives
	L-Serine	C3H7NO3	epd00054	A1, C2	Amino Acids and Derivatives
	Urea	CH4N2O	epd00073	P1, C2	Organic Acids

P1	Ornithine	C5H13N2O2	cpd00064	P1	Nucleosides/Nucleotides and Derivatives
	Adenosine	C10H13N5O4	cpd00182	B4	Nucleosides/Nucleotides and Derivatives
	L-Tyrosine	C9H11NO3	cpd00069	C2	Amino Acids and Derivatives
	Cytidine	C9H13N3O5	cpd00367	B4	Nucleosides/Nucleotides and Derivatives
	Uracil	C4H4N2O2	cpd00092	A1, L3	Nucleosides/Nucleotides and Derivatives
	L-Phenylalanine	C9H11NO2	cpd00066	A1, C2, S4	Amino Acids and Derivatives
	Myristic acid	C14H27O2	cpd03847	L3, B4, S4	Organic Acids
	Hypoxanthine	C5H4N4O	cpd00226	C2, L3, S4	Nucleosides/Nucleotides and Derivatives
	Glycerol	C3H8O3	cpd00100	L3	Saccharides and Derivatives
	D-Fructose	C6H12O6	cpd00082	A1, C2	Saccharides and Derivatives
	Succinate	C4H4O4	cpd00036	C2, S4	Organic Acids
	Deoxycytidine	C9H13N3O4	cpd00654	B4	Nucleosides/Nucleotides and Derivatives
	D-Trehalose	C12H22O11	cpd00794	S4	Saccharides and Derivatives
	D-Alanine	C3H7NO2	cpd00117	A1, C2	Amino Acids and Derivatives
	L-Serine	C3H7NO3	cpd00054	C2	Amino Acids and Derivatives
	S4	L-Glutamate	C5H8NO4	cpd00023	A1, C2, L3, B4, S4
L-Arginine		C6H15N4O2	cpd00051	L3	Amino Acids and Derivatives
D-Mannose		C6H12O6	cpd00138	A1, C2	Saccharides and Derivatives
Nicotinamide ribonucleotide		C11H14N2O8P	cpd00355	P1	Vitamins
Palmitate		C16H31O2	cpd00214	A1	Organic Acids
(R)-3-Hydroxybutanoate		C4H7O3	cpd00797	A1	Organic Acids
N-Acetyl-D-glucosamine		C8H15NO6	cpd00122	P1	Saccharides and Derivatives
D-Mannose		C6H12O6	cpd00138	A1, C2	Saccharides and Derivatives

”

6. General Modeling Methodology

Lines 70–82

• The introduction describes the novelty of applying GSMM to co-contamination, but doesn’t discuss possible pitfalls (e.g., lack of spatial constraints, dynamic succession). In addition, it would be beneficial to know what the most suitable constraint-based modeling approaches are to use under these types of scenarios. Would you add 1–2 sentences acknowledging these modeling limitations and some modeling preludes to this work?

Answer: Thanks for the reviewer’s comments. We agree that the introduction should acknowledge the limitations of GSMMs in co-contamination scenarios. In response, we have revised the manuscript to highlight a key challenge: the difficulty in systematically exploring all possible strain combinations and balancing their cooperative and competitive interactions. We further indicate that constraint-based approaches capable of navigating this vast design space without predefining interaction types are needed. Our Super Community Combinations (SuperCC) framework is introduced as a prelude to address this exact gap, enabling comprehensive analysis without restrictions on community size.

On Lines 69-79, we write:

“...Additionally, it is important to note that a key limitation in applying GSMMs to multi-strain communities lies in the difficulty of systematically exploring all possible strain combinations and balancing their cooperative and competitive interactions. To address this, constraint-based approaches that can navigate the vast design space of complex communities without predefining interaction types are needed. Currently, no systematic framework exists for the GSMM-guided design of synthetic microbiomes under multi-pollutant stress, particularly one that integrates native microbial functionalities with engineered community-level stability. To bridge this gap, we introduce our Super Community Combinations (SuperCC) framework as a prelude, which is designed to address this exact need by enabling such comprehensive analysis without limitations on microbiome

size.

In this study, we demonstrate how the SuperCC framework enables the systematic design of synthetic microbiomes for co-contamination. Specifically, we..."

7. Experimental Design Controls

Lines 217–239, 474–490

Did you run abiotic controls (pollutants in medium without microbes) to measure any chemical degradation independent of biological activity? If so, where are these data shown?

Answer: Thanks for the reviewer's comments. Yes, abiotic controls (containing the respective pollutants in sterile medium without microbial inoculation) were systematically included in every degradation experiment throughout our study. This was implemented for both the initial microbial consortium acclimation phases and the subsequent assays evaluating the degradation efficacy of individual keystone strains.

The data from these abiotic controls are explicitly shown in Supplementary Figure S4, specifically represented by the groups labeled as "CK". These control measurements confirm that the chemical degradation observed in our study is attributable to biological activity rather than non-biological processes.

On Fig. S4, we write:

“

Fig. S4 | Degradation capacity of six keystone strains against TC and OTC. Since none of these strains could utilize the pollutants as sole carbon sources, degradation tests were conducted in minimal medium (MM) supplemented with 10% peptone, containing either 30 mg/L TC or OTC for 3 days' cultivation. *, $p < 0.05$; **, $p < 0.01$; ***, $p < 0.001$. TC, tetracycline; OTC, oxytetracycline. CK, control group. The data are presented as mean values \pm SD ($n = 3$ biological independent replicates). ”

Were non-acclimated consortia tested under the same pollutant stress to isolate the effect of the acclimation process?

Answer: Thanks for the reviewer's comments. We thank the reviewer for raising this valuable point, which indeed improves the rigor of our study. We acknowledge that including a parallel control of a completely non-acclimated community under the same pollutant stress was not part of our experimental design, which we recognize as a limitation and have noted in the Discussion section. The original objective of this study was to enrich and isolate tetracycline-degrading bacteria, with particular interest in multifunctional strains capable of degrading both TC and OTC. Using a gradient

domestication strategy, we exposed microbial communities to both antibiotics and monitored residual concentrations. Interestingly, we observed that co-contamination-adapted consortia demonstrated enhanced degradation capacity and increased degrader diversity—a finding consistent with reports by Smith et al. on enhanced microbial adaptation under combined stress. This unanticipated result shifted our focus toward investigating the potential degradation advantages under co-contamination conditions. Consequently, systematic analysis of the initial microbial community's degradation capacity and compositional changes during early acclimation was not performed.

We are currently conducting a follow-up study to address this specific point. In this experiment, we collected farmland soil samples with long-term tetracycline contamination as well as pristine, uncontaminated soil. These were subjected to both single-pollutant and co-contamination acclimation conditions. During the initial acclimation phase (first week), all microbial consortia exhibited relatively weak degradation capacity. However, after one month of acclimation, the degradation performance of consortia derived from pre-contaminated soil improved significantly and substantially outperformed those from the pristine soil. These results further underscore the critical importance of the acclimation process.

In the metabolite supplementation experiments, could growth effects be due simply to extra C/N input rather than specific metabolites?

Answer: Thanks for the reviewer's valuable comments. According to your suggestion, we conducted additional experiments. Under the same conditions (inorganic salt medium containing 10 % peptone, tetracycline 10 ppm, and oxytetracycline 10 ppm), we supplemented glucose at equivalent or double the carbon molar amount of the added metabolites. The results showed that doubling the glucose input did not result in a proportional increase in growth (Fig. S10b, c). In contrast, supplementation with AA promoted microbial growth under combined antibiotic stress more effectively than adding twice the amount of glucose (Fig. S10a).

On Lines 306-308, we write:

"...We also verified that the growth-promoting effect of metabolite supplementation stems from the specific metabolites themselves, rather than the additional carbon input (Fig. S10)."

On Fig. S10, we write:

"

Fig. S10 | Equivalent glucose supplementation experiments. Control experiments were conducted to assess whether the growth-promoting effects of metabolite supplementation were attributable to additional carbon input rather than the specific metabolites themselves. Growth curves of microbial consortia under TC and OTC co-contamination with supplementation of (a) amino acids (AA), (b) vitamins (Vit), or (c) organic acids (OA) were compared with glucose supplementation providing an equivalent or double molar amount of carbon. Each panel shows three treatments: addition of the corresponding metabolite (AA, Vit, or OA), glucose at the same molar amount of carbon as the metabolite, and glucose at twice the molar amount of carbon. Growth was monitored by optical density at 600 nm over 2400 min. TC, tetracycline; OTC, oxytetracycline. The data are presented as mean values ($n = 4$ biological independent replicates).”

8. Statistical Robustness

Lines 410–429, 162–189

Were multiple testing corrections applied to metabolomics p -values? If so, which method? If not, would you consider applying FDR correction given the high number of features?

How many biological replicates (n) were used for each metabolomics condition? It would be helpful to state this explicitly in the figure legend for Fig. 4.

Answer: We sincerely thank the reviewer for this thoughtful comment on statistical robustness. In line with the suggestion, we also applied FDR correction to identify significantly altered metabolites. The results showed that applying either p -value < 0.05 , VIP score ≥ 1 , and fold change ≥ 1 , or FDR < 0.05 , VIP score ≥ 1 , and fold change ≥ 1 produced very similar outcomes. The corresponding results have been added to the Supplementary Table S3. For clarity and consistency, we therefore present the results based on p -values in the main text. Each experimental condition included $n = 4$ independent biological replicates. We have now explicitly stated this in the legend of Fig. 4. In addition, the Methods section has been revised to provide more detailed information on the design

and implementation of the replicate experiments.

On Lines 757-758, we write:

“...Data are presented as mean ± SD; n = 4 biological replicates...”

On Table S3, we write:

“*Supplementary Table S3 Statistics of differential metabolite screening under different statistical criteria.*”

Group	Screening criteria for differential metabolites					
	p-value < 0.05, VIP score ≥ 1, and fold change ≥ 1			FRD < 0.05, VIP score ≥ 1, and fold change ≥ 1		
	Number of up-regulated metabolites	Number of down-regulated metabolites	All	Number of up-regulated metabolites	Number of down-regulated metabolites	All
L_TO vs L_CK	559	316	875	552	306	858
B_TO vs B_CK	510	177	687	506	177	683
S_TO vs S_CK	507	163	670	506	159	665
P_TO vs P_CK	498	165	663	495	164	659
A_TO vs A_CK	543	154	697	537	147	684
C_TO vs C_CK	506	130	636	503	128	631
AP_TO vs AP_CK	586	191	777	584	187	771
PB_TO vs PB_CK	582	197	779	579	193	772
AS_TO vs AS_CK	598	152	750	593	148	741
PC_TO vs PC_CK	545	157	702	540	147	687
PS_TO vs PS_CK	535	190	725	533	186	719
CB_TO vs CB_CK	597	172	769	588	168	756
AB_TO vs AB_CK	617	109	726	615	107	722
SB_TO vs SB_CK	522	169	691	522	168	690
CS_TO vs CS_CK	539	145	684	534	143	677
AC_TO vs AC_CK	583	129	712	576	123	699
CSB_TO vs CSB_CK	555	219	774	548	193	741
PCB_TO vs PCB_CK	608	168	776	600	163	763
ACS_TO vs ACS_CK	622	141	763	612	136	748
FSB_TO vs FSB_CK	566	164	730	564	161	725
APC_TO vs APC_CK	762	239	1001	759	237	996
APB_TO vs APB_CK	619	198	817	615	192	807
ACB_TO vs ACB_CK	633	134	767	632	134	766
APS_TO vs APS_CK	598	159	757	589	157	746
PCS_TO vs PCS_CK	542	169	711	540	162	702
ASB_TO vs ASB_CK	603	104	707	573	86	659

Note: TO: Treatment group exposed to a combination of tetracycline and oxytetracycline. CK: Control group without pollutant exposure. The letters (e.g., L, B, S) denote different bacterial consortia: L3, B4, S4, A1, C2, P1.
”

9. Mechanistic Evidence for Potentiators

Lines 191–215, 316–336

Are there any partial removal experiments or mutant strain data for B4 and S4 that could directly test the potentiator role?

Would you agree that potentiator function in your study is inferred rather than experimentally proven? Please add a cautionary note, making this explicit.

Answer: Thanks for the reviewer’s comments. The growth and metabolism of the 6-strain consortium and the 4-strain consortium with B4 and S4 removed are shown in Figure 6b, from which the roles of B4 and S4 can be observed. To further clarify the quantitative definitions of degrader, helper, and especially potentiator, we introduce a Potentiator Contribution Index (PCI) to quantitatively measure the potentiator’s effect across conditions. We have incorporated this information into Table 1 and calculated the PCI values for microbial growth and degradation rate in Table S7, all of which highlight the importance of the Potentiator.

Initially, our understanding of the potentiator's function was derived from speculation and model simulation. However, this conjecture was subsequently supported by experimental results such as metabolomics and strain degradation ability verification, emphasizing the unheralded contribution of

such bacteria in the actual environment. This experiment did not involve mutant strain verification experiments because this auxiliary function is reflected in many aspects, not only the promotion of degradation but also the promotion of growth and metabolism. Therefore, there is no way to obtain mutant strains for experimental verification, and there is no need for mutation experiments, because the verification design in Figure 6 and the PCI values can already illustrate their contribution.

On Table 1, we write:

“**Table 1** Characteristics of Degraders, Helper, Potentiator, and Potentiator Contribution Index (PCI) in DHP-Com.

	Abundance	Significant difference in changes	Function	In this study
Degrader	Not necessarily high	Yes	Direct function, degrade pollutants	Strain C2
Helper	Not necessarily high	Yes	Indirect function, help degraders	Strains A1, P1, L3
Potentiator	High	No	Indirect function, help degraders and helpers	Strains B4, S4 (The abundance is usually among the top 30.)
Potentiator Contribution Index (PCI)	$PCI = \frac{\text{Performance (DHP)} - \text{Performance (DH)}}{\text{Performance (DH)}} \times 100\%$			

Performance (DHP) = Overall performance (e.g., pollutant degradation efficiency, community stability) of the consortium containing *Degrader* (D), *Helper* (H), and *Potentiator* (P).

Performance (DH) = Performance of the consortium with only *Degrader* (D) and *Helper* (H). „

On Lines 85-86, we write:

“...and also introducing a Potentiator Contribution Index (PCI) to quantitatively measure the potentiator’s effect across conditions (Table 1).”

On Table S7, we write:

“**Table S7** Potentiator Contribution Index (PCI) for Degradation Rate and Biomass under Different Pollutants.

Pollutants	PCI (degradation rate)	PCI (biomass)
DTAC	69.51886277	6.461001164
DBAC	79.39711011	
SM2	45.66666667	31.25
SMX	-23.24324324	
TC	394.9874687	37.88235294
OTC	96.28081906	
BO	10.78869859	32.18218218
QC	-0.463320463	

„

10. Choice of GEM Reconstruction Tool

Lines 431–444

Provide rationale for using ModelSEED over alternatives (CarveMe, gapseq, AGORA), detail manual curation steps, gap-filling criteria, and whether benchmarking against existing GEMs was performed.

Answer: Thanks for the reviewer’s comments. Compared with other modeling software, ModelSEED has the following advantages: (1) Higher model accuracy and detail completeness: It adopts a bottom-up construction approach, integrates multi-source data, and includes thermodynamic

constraints and compartmentalization information, enabling it to more truly reflect the complexity of biological metabolic networks; (2) Data standardization and comprehensiveness: Its database has undergone standardization processing and contains details such as transport reactions and charged molecule balance, providing a foundation for accurate Flux Balance Analysis (FBA); (3) Community-driven continuous optimization: Through a community maintenance mechanism, it is continuously updated and improved to ensure the timeliness and expandability of data, while supporting user contributions and modifications; (4) Flexibility and adaptability: It is applicable to a variety of organism types and research scenarios, and performs exceptionally well especially in studies that require in-depth analysis of metabolic network details—unlike AGORA, which is limited to gut microbiota; (5) Direct functional usability: ModelSEED can directly output models usable for FBA, eliminating the need for additional format conversion or processing steps (see the table below for details).

Therefore, considering the aforementioned advantages and its better compatibility with the SuperCC tool for subsequent analyses, we ultimately selected ModelSEED as the tool for constructing single-strain model drafts.

	ModelSEED	CarveMe	gapseq	AGORA
Core Approach	Bottom-up (multi-source reactions+optimization)	Top-down (BiGG-based universal model + screening)	Semi-bottom-up (ModelSEED + flux analysis)	Comparative genomics (human gut focus)
Key Strengths	High accuracy ; comprehensive data; FBA-ready	Fast; no extra annotations; large-scale modeling	Precise enzyme/carbon source prediction; multi-db integration	Specialized for gut microbiota; drug metabolism focus
Key Limitations	Complex; high computing needs	Misses species-specific pathways; lower accuracy	Poor non-bacterial support; needs expertise	Narrow scope (only gut)
Best For	Metabolic engineering; detailed network study	Rapid microbiome screening; exploratory research	Bacterial physiology; fermentation optimization	Gut-host interaction; gut-drug studies

After obtaining the single-strain model drafts, we subjected each draft to a rigorous manual curation process following the workflow described in two articles, including curation using COBRAToolbox-3.0:

1. Henry, C., DeJongh, M., Best, A. et al. High-throughput generation, optimization and analysis of genome-scale metabolic models. *Nat Biotechnol* 28, 977–982 (2010).
2. Thiele, I., Palsson, B. A protocol for generating a high-quality genome-scale metabolic reconstruction. *Nat Protoc* 5, 93–121 (2010).

Both of the above two literatures have been newly inserted into the manuscript, at Line 453.

We tested the growth of each strain in minimal medium (MM) with different carbon sources and these experimental results guided the model curation. For each strain, we performed growth

simulations using at least three different carbon sources as the sole carbon source, leveraging the experimental findings to supplement missing metabolic reactions. Generally, the draft models failed to produce all biomass components under specific nutrient conditions where experimental evidence confirmed that strain growth was feasible.

Potentially missing reactions were first identified through an automated gap-filling process, with only those supported by gene evidence in either the strain's own genome or the genomes of its phylogenetically close relatives being retained. We then manually supplemented missing reactions based on gene and enzyme annotations from public databases including KEGG, UniProt, BiGG, IMG, and MetaCyc.

In addition, we standardized reaction IDs across different databases for consistency, checked and balanced elementally imbalanced reactions based on chemical formulas, and removed futile loops. Each reversible reaction involved in CO₂ production and utilization was manually corrected, and the single-strain model of each bacterium was rigorously curated using the same standardized approach. Following iterative revisions, the final models were able to produce all biomass components in minimal medium with various carbon and nitrogen sources, consistent with experimental results.

Additional Reproducibility and Data/Code Availability

Please indicate where the data and code used in this study — including the SuperCC modeling scripts, genome-scale metabolic models, simulation parameters, and metabolomics datasets — can be accessed to reproduce the modeling and simulation results and to retrieve the processed and raw metabolomics data. If available, provide persistent links (e.g., GitHub, Zenodo, Figshare) and accession numbers for public repositories, along with any instructions necessary for execution.

Answer: Thanks for the reviewer's comments. The code and models used for the analysis is available on GitHub (<https://github.com/ruanzhepu/superCC>), and the "Code Availability" section has been supplemented in the manuscript. The metabolomics data have been deposited in the MetaboLights database under the accession number MTBLS12809. This information has also been added to the Data Availability section.

MINOR COMMENTS

- Define “potentiator” in Abstract for broader readership.

Answer: Thanks for the reviewer's comments. To make the term "potentiator" accessible to a broader readership, we have supplemented its definition in the Abstract.

On Lines 31-32, we write:

“...DHP-Com (Degradar-Helper-Potentiator). Potentiators are top species with stable habitat abundance...”

- Clarify biomass coefficient weighting (line 470).

Answer: Thanks for the reviewer's comments. In our SuperCC modeling tool, we offer four scenario options for setting the biomass weighting of community models, as follows:

(1) Scenario 1: The objective function of the community model is the sum of the biomass of all strains, with no restrictions on the biomass proportion of each strain (allowing a strain's biomass to be 0).

- (2) Scenario 2: The biomass of the first strain is used as the objective function of the community model, which is suitable for screening other strains that can promote the biomass of this first strain.
- (3) Scenario 3: The objective function is the sum of the biomass of all strains, with the constraint that the biomass proportion of all strains is 1 (i.e., equal proportions).
- (4) Scenario 4: The objective function is the sum of the biomass of all strains, and the biomass proportion of each strain is determined by the artificially set “*PercentOfSpeciesBio*” parameter (this parameter must be consistent with the number of strains and all values must be positive).

These scenario designs can flexibly meet the needs of simulating and analyzing biomass distribution in community models under different research requirements.

On Lines 496-500, we write:

“Four commonly used scenarios were provided, including 1) equal abundance for each organism; 2) no limitation for any organisms, meaning that the biomass of each organism is allowed to be zero; 3) defining biomass of a specific organism as community biomass (used for identifying organisms in a community that could improve the growth of the target organism); and 4) any defined abundances.”

- Improve figure readability (Figs. 2i, 4, 5c, 6) and include complete metabolite class legends.

Answer: Thanks for the reviewer’s valuable feedback. We have thoroughly revised and improved the readability of Figures 2i, 4, 5c, and 6, including the addition of complete metabolite class legends. The updated figures are presented below for your review.

On Fig. 2i, we write:

“i, Topological analysis of species co-occurrence networks among the Top-50 species across all three acclimation conditions. Nodes represent species (ASVs), with node size proportional to its degree (number of connections). Edges represent statistically significant strong correlations ($|r| \geq 0.9$, $p \leq 0.05$) between species, with their color corresponding to the abundance rank category of the connecting nodes. The table below the network panels provides a summary of the key topological parameters for each of the three scenarios...”

On Fig. 4, we write:

“

...**b**, Quantitative analysis of significantly upregulated metabolites (compared to CK; $p < 0.05$, $VIP > 1$ in OPLS-DA), displaying metabolite counts (represented by color-coded circles) and mean fold-change ($\log_2FC \pm SEM$). The microbial consortium abbreviations (APCLSB) in the figure correspond to strains A1, P1, C2, L3, B4, and S4, respectively. **c**, Functional characterization of metabolites under combined antibiotic stress (TCs). Top: Venn diagram and functional annotation of TCs-associated metabolites. Small nodes represent individual metabolites, while yellow and grey circles indicate metabolites shared among or unique to different microbial communities, respectively. Connecting lines represent predicted metabolite production capability. Circular sectors display Human Metabolome Database (HMDB)-annotated metabolite classes, with central numbers indicating the total count of annotated metabolites. Center: Circular annotation plot specifically highlights metabolite distribution patterns. The grey ring displays metabolites common to all three community types; the yellow ring represents metabolites shared between single- and tri-strain communities; the blue ring denotes metabolites shared between dual- and tri-strain communities; the light red ring highlights metabolites unique to the tri-strain consortium. Bottom: Adjacent lollipop plots depict key metabolites predicted by random forest analysis (Mean Decrease Accuracy; $p < 0.01$, $VIP > 2$), colored according to their membership in the categories outlined above (yellow/blue/light red)...

On Fig. 5c, we write:

“c, Flux balance analysis illustrating interspecies metabolic exchange under the objective of maximizing total community biomass. Each node (large circle) represents a bacterial strain, with directed arrows indicating the direction of metabolite transfer. Color-coded peripheral symbols (small circles) represent metabolite classes, and numeric labels indicate the specific number of metabolite types exchanged within each class. The annotation in the upper left corner shows the number of carbon sources available in the medium and the simulation time (20 hours) required for the microbial community to reach optimal growth. Grey lines represent unidirectional exchange reactions originating exclusively from strain B4.”

On Fig. 6, we write:

“**Fig. 6** | Experimental validation of predicted metabolite-mediated interactions under co-contaminated conditions. a, Growth promotion by metabolite supplementation. Growth curves illustrate the responses of microbial communities to four types of combined pollution (10 mg/L each), including tetracyclines (TC and OTC), sulfonamide antibiotics (SMX and SMT), herbicides (BO and QC), and biocides (DTAC and DBAC). Various metabolite combinations were added to the culture systems and monitored at 15-minute intervals for 1,800 minutes at 30°C using a microplate reader, resulting in 9,680 growth measurement data points per treatment. The control group (Com) was cultured in basal medium (containing 10% peptone), while the experimental groups were supplemented with additional amino acids (Com+AA: L-histidine, L-glutamate, L-phenylalanine, N-formyl-L-methionine, glycine, and L-proline), vitamins (Com+Vit: vitamin B1, vitamin D2, and niacin), organic acids (Com+OA: palmitate, myristic acid, and succinate), or a mixture of all key metabolites (Com+All). The top-left inset bar plot quantifies the area under each growth curve. Data are presented as mean \pm SD ($n = 4$ biologically independent replicates). b, Validation of the altruistic effects of strains B4 and S4 (including promoting microbial community growth and pollutant removal). Bar plots represent the removal efficiencies of mixed pollutants by three microbial consortia (left Y-axis). Lowercase letters above the bars indicate statistically significant differences between groups (one-way ANOVA, $p < 0.05$), where black letters represent differences among pollutants in orange color blocks, and gray letters represent differences among pollutants corresponding to green color blocks. Line plots represent bacterial growth (measured as absorbance at OD600; right Y-axis). ** $p < 0.01$, *** $p < 0.001$. The tested consortia were: BS (strains B4 and S4), BS+ACLP (strains B4, S4, A1, C2, L3, P1), and ACPL (strains A1, C2, L3, P1). Data are presented as mean \pm SD ($n = 3$ biologically independent replicates). TC, tetracycline; OTC, oxytetracycline; DTAC, dodecyl trimethyl ammonium chloride; DBAC, dodecyl dimethyl benzyl ammonium chloride; BO, bromoxynil octanoate; Bxn, bromoxynil (intermediate metabolite of BO); QC, quinclorac; SMX, sulfamethoxazole; SMT, sulfamethazine.”

- Standardize “co-contamination” vs “co-pollutant” terminology.

Answer: Thanks for the reviewer’s comments. To address the terminology consistency, we have standardized the relevant expression throughout the manuscript to "co-contamination".

On Lines 502-510, we write:

“Verification of key metabolites' role on co-contamination remediation

...The consortium was inoculated into MM containing 10% (w/v) peptone along with four distinct co-contaminated mixtures...(1) Com: Microbial consortium + 10 mg/L two pollutants...”

- Include full media compositions in Supplementary.

Answer: Thanks for the reviewer’s comments. We have included the complete compositions of all media used in the strain metabolic models and simulations as attachments in the "Supplementary models" folder. Additionally, we have uploaded this relevant content to the GitHub repository (available at <https://github.com/ruanzhepu/superCC.git>) for easy access and reference.

Recommendation: Strong and methodologically advanced manuscript. However, I recommend major revisions to this manuscript to address several shortcomings. After a major overhaul, the manuscript will be significantly improved.

Answer: We sincerely thank the reviewer for the encouraging overall assessment and for the constructive suggestions provided. We fully acknowledge the shortcomings identified and deeply appreciate the valuable guidance. We are committed to carefully addressing all comments through substantial revisions and improvements to the manuscript. We believe that implementing these changes will significantly enhance the clarity, rigor, and overall quality of our work.

Reviewer #2 (Remarks to the Author):

In this manuscript, the authors use a structured approach for targeted microbiome application by integrating top-down and bottom-up methods. This approach involves domesticating bacterial communities from natural water samples and leveraging in-silico strategies to optimize both the functional performance and stability of the resulting synthetic microbiomes. They design different treatment groups: adding a single antibiotic (either tetracycline or oxytetracycline) or co-adding both antibiotics. These treatments aim to analyze and identify keystone species in water samples, as well as enable directional enrichment of microbial communities in multi-pollutant contaminated environments. The authors further employ significance analysis to confirm these keystone species and use a compartmentalized community SuperCC approach—a powerful synthetic microbial community design framework independently developed by the study's authors—to evaluate metabolic flux exchange among different members of the microbial community.

The authors' study is quite interesting: they isolated six keystone species from polluted water, and all of these species align perfectly with the three functional roles proposed by the authors. Unlike the traditional synthetic microbial community model of "degrader bacteria + helper bacteria," the authors proposed a third bacterial type termed "Potentiators" and highlighted their pivotal role within the microbial community. The discovery and proposal of Potentiators have addressed the long-standing challenge in practical applications where microbial communities exhibit poor colonization and high mortality. These previously overlooked, inconspicuous bacteria play a crucial role in assisting functional microbial communities in colonization and degradation processes. To validate this hypothesis, the authors used the SuperCC genome-scale metabolic model to make predictions. By using different types and quantities of carbon sources to simulate single and composite pollution scenarios, they have significantly overcome the predicament where metabolic model simulations are hindered by the difficulty in clarifying microbial degradation pathways of pollutants in environments with emerging or composite pollutants. This approach offers theoretical and technical support for future simulations of microbial community interactions in complex polluted environments.

For the proposed DHP-Com framework, the authors further selected 8 organic pollutants belonging to 4 categories. Through this approach, they successfully enabled the microbial consortium to achieve efficient degradation capacity, as well as enhanced tolerance and growth capabilities. This work paves the way for broad application prospects in enhancing the adaptability and functionality of synthetic microbial communities across diverse complex environments. Overall, this is a potentially impactful paper.

Answer: We sincerely thank Reviewer 2 for the positive and encouraging comments on our work. We truly appreciate your recognition of the study's significance and contributions. We have carefully addressed all the specific suggestions regarding article formatting, visual presentation, and standardization, as detailed in our point-by-point responses below.

Smaller comments:

-The format of references needs to be further confirmed. In some references, the species names in the titles are not italicized.

Answer: Thanks for the reviewer's comments. We have carefully checked and standardized the

formatting of all references, with particular attention to italicizing the words that require it.

-All sequencing data in the manuscript (including raw data from metabolomic analyses) should be uploaded to public database platforms.

Answer: Thanks for the reviewer's comments. The metabolomics data have been deposited in the MetaboLights database under the accession number MTBLS12809. This information has also been added to the Data Availability section.

-Lines 465-466, Fix the line-breaking format of the formula.

Answer: Thanks for the reviewer's comments. We have checked and corrected the format of the formula.

-The formatting of P-values in the manuscript appears inconsistent. Additionally, a space should be inserted between "P" and "<". Please recheck and correct this throughout the entire manuscript.

Answer: Thanks for the reviewer's comments. We have thoroughly rechecked and corrected the formatting of *p*-values throughout the entire manuscript and Supplementary Information. Specifically, we standardized all instances to use lowercase and italicized "*p*" and added the required space between "*p*" and the symbol.

-Lines 49-50 and 64-65 lack literature support. Please add relevant references and conduct a full-manuscript check.

Comments on Figures:

Answer: Thanks for the reviewer's comments. We have supplemented the manuscript with supporting reference as follows:

Reference [7] (Ruan, Z. et al. Engineering natural microbiomes toward enhanced bioremediation by microbiome modeling. *Nat Commun* 15, 4694 (2024)) has been added to Line 46.

Comments on Figures:

-Figure 1 could be appropriately enlarged, especially Figure 1a. As it presents the experimental framework and new concepts, the current size of the figure is somewhat small, which may affect readability.

Answer: Thanks for the reviewer's comments. We have not only adjusted the overall size of Figure 1a but also supplemented a more detailed analysis workflow based on the storyline of the results, thereby enhancing the coherence between Figure 1a and the other main figures.

On Fig. 1, we write:

“

Fig. 1 | Experimental design and microbial consortium responses to combined antibiotic pollution. *a*, The workflow involves the domestication of aquatic microbial communities from the same water sample using three acclimation approaches: exposure to tetracycline (TC) alone, oxytetracycline (OTC) alone, or a combination of both (TCs). Functional microbiomes were subsequently obtained and analyzed via amplicon sequencing. Meanwhile, systematic analysis was conducted on the metabolic models of the top 50 strains in terms of abundance in each treatment group to determine the growth and metabolic characteristics of the dominant species. Potential keystone species were thus identified through an integrated approach combining abundance shifts and strain isolation. To assess strain-specific metabolic responses under varying stress conditions, untargeted metabolomics was conducted on single-, co-, and tri-strain cocultures under three treatments: antibiotic-free control (CK), exposure to a single antibiotic (TC or OTC), and exposure to the combined TCs treatment. Since metabolomic analysis was only performed on co-cultures with up to three strains, the microbial community modeling framework SuperCC was applied to construct multi-strain metabolic models with a larger number of strains, aiming to predict strain-specific biomass and identify key interspecies metabolites. Model predictions were subsequently validated through experimental analyses. Based on the above conclusions, we proposed a synthetic microbiome construction strategy: the Degradер–Helper–Potentiator Consortium (DHP-Com) paradigm, and provided the definition of “potentiator”...”

-The line spacing between each row of the random forest results below Figure 4C is inconsistent.
Answer: Thank you for your comments. We have revised the random forest plot in the lower part of Figure 4C. In addition to standardizing the consistent line spacing between each row, we have also added the classification information for these metabolites.

On Fig. 4, we write:
 “

...**b**, Quantitative analysis of significantly upregulated metabolites (compared to CK; $p < 0.05$, $VIP > 1$ in OPLS-DA), displaying metabolite counts (represented by color-coded circles) and mean fold-change ($\log_2FC \pm SEM$). The microbial consortium abbreviations (APCLSB) in the figure correspond to strains A1, P1, C2, L3, B4, and S4, respectively. **c**, Functional characterization of metabolites under combined antibiotic stress (TCs). Top: Venn diagram and functional annotation of TCs-associated metabolites. Small nodes represent individual metabolites, while yellow and grey circles indicate metabolites shared among or unique to different microbial communities, respectively. Connecting lines represent predicted metabolite production capability. Circular sectors display Human Metabolome Database (HMDB)-annotated metabolite classes, with central numbers indicating the total count of annotated metabolites. Center: Circular annotation plot specifically highlights metabolite distribution patterns. The grey ring displays metabolites common to all three community types; the yellow ring represents metabolites shared between single- and tri-strain communities; the blue ring denotes metabolites shared between dual- and tri-strain communities; the light red ring highlights metabolites unique to the tri-strain consortium. Bottom: Adjacent lollipop plots depict key metabolites predicted by random forest analysis (Mean Decrease Accuracy; $p < 0.01$, $VIP > 2$), colored according to their membership in the categories outlined above (yellow/blue/light red)...

-It is recommended to rename the y-axis label of Figure 6b to "degradation rate (%)".

Answer: Thanks for the reviewer's comments. We have modified the y-axis label.

Reviewer #3 (Remarks to the Author):

SUMMARY

I found the manuscript intriguing and potentially impactful, but I had to re-read multiple sections due to organizational and definitional ambiguity. The scientific core feels strongest as a methodological advance—extending multi-strain GSMM/FBA to guide community design, while the specific strain findings are illustrative rather than the central contribution. Clarity around role definitions, statistical support for several comparative claims, and figure/text alignment would materially improve readability and confidence.

NOTEWORTHY RESULTS

- Model and bench alignment: Multistrain GSMM/FBA predictions are borne out experimentally, supporting the feasibility of model-guided, role-based consortium design.
- Mechanistic thread (vitamin rescue): Supplementation patterns (namely vitamins) directionally support a cross-feeding mechanism consistent with omics signals.
- Field impact: The work strengthens a known ecological principle (multi-member robustness) but advances the analysis toolkit for environmental biotech and microbial ecology by making >2-member modeling actionable.

SIGNIFICANCE

The ecological idea (in essence, that more species of the "potentiator" class can translate to greater performance/robustness, as I understood it) is not wholly novel per se, but formalizing it with genome-based multistrain modeling and using that to nominate compositions/roles feels like a meaningful step forward. The contribution might be more primarily one of method/analysis, not a new ecological law.

Answer: We sincerely thank the reviewer for the constructive feedback and recognition of the methodological advance of extending multi-strain GSMM/FBA to guide community design. We fully acknowledge the concerns regarding clarity, role definitions, statistical support, and figure/text alignment, and we will revise the manuscript to improve readability and rigor. We also agree with the reviewer's perspective that our main contribution lies in the methodological framework, while ecological findings serve as illustrative demonstrations. We greatly appreciate these suggestions, which will help us strengthen the clarity, rigor, and impact of the work.

SUPPORT OF CLAIMS/CONCLUSIONS

- Some comparative claims might make use of additional statistics. Phrases such as “contrasting patterns” between pollutant groups should be backed by named tests, effect sizes, and multiple-testing corrections.

Answer: We thank the reviewer for this insightful comment. To ensure the robustness of our comparative statements, we have specified the statistical analyses used to support them.

On Lines 96-98, we write:

“...showed superior degradation efficiency of tetracycline antibiotics, particularly OTC, compared to the single-pollutant groups (one-way ANOVA, $p = 0.00098$; Fig. 1b).”

On Lines 100-101, we write:

“...the TCs consortium exhibited significantly higher α -diversity (one-way ANOVA, $p = 0.03411$).”

- Please provide explicit quantitative definitions for degrader/helper/potentiator, especially potentiator. I recommend introducing and reporting a potentiator index metric/score of some kind across conditions that will facilitate comparison/ranking. Perhaps: [Performance(DH+P) - Performance(DH)] / Performance(DH). Where P = Potentiator and DH = Degrader+Helper.

Answer: Thanks for the reviewer’s comments. We appreciate the reviewer’s constructive suggestion to clarify the quantitative definitions of degrader, helper, and especially potentiator. To address this, we introduce a Potentiator Contribution Index (PCI) to quantitatively measure the potentiator’s effect across conditions. We have incorporated this information into Table 1 and calculated the PCI values for microbial growth and degradation rate in Table S7, all of which highlight the importance of the Potentiator.

On Table 1, we write:

“**Table 1** Characteristics of Degrader, Helper, Potentiator, and Potentiator Contribution Index (PCI) in DHP-Com.

	Abundance	Significant difference in changes	Function	In this study
Degrader	Not necessarily high	Yes	Direct function, degrade pollutants	Strain C2
Helper	Not necessarily high	Yes	Indirect function, help degraders	Strains A1, P1, L3
Potentiator	High	No	Indirect function, help degraders and helpers	Strains B4 , S4 (The abundance is usually among the top 30.)
Potentiator Contribution Index (PCI)	$PCI = \frac{\text{Performance (DHP)} - \text{Performance (DH)}}{\text{Performance (DH)}} \times 100\%$			

Performance (DHP) = Overall performance (e.g., pollutant degradation efficiency, community stability) of the consortium containing Degrader (D), Helper (H), and Potentiator (P).

Performance (DH) = Performance of the consortium with only Degrader (D) and Helper (H). ”

On Lines 85-86, we write:

“...and also introducing a Potentiator Contribution Index (PCI) to quantitatively measure the potentiator’s effect across conditions (Table 1).”

On Table S7, we write:

“**Table S7** Potentiator Contribution Index (PCI) for Degradation Rate and Biomass under Different Pollutants.

Pollutants	PCI (degradation rate)	PCI (biomass)
DTAC	69.51886277	6.461001164
DBAC	79.39711011	
SM2	45.66666667	31.25
SMX	-23.24324324	
TC	394.9874687	37.88235294
OTC	96.28081906	
BO	10.78869859	32.18218218
QC	-0.463320463	

- "Ablation" test: For the best trio, experimentally validate DH vs DH+P and swap P for a near-neighbor to demonstrate role specificity rather than a generic third-member effect.

Answer: Thanks for the reviewer’s valuable comment. To further validate the specificity of the enhancing effect, we conducted an additional experiment using the degrader *Comamonas* sp. C2 (C2), the helper strain *Providencia* sp. P1 (P1), and the enhancer strain *Brevundimonas* sp. B4 (B4), together with two closely related *Brevundimonas* strains (B-1 and B-2). The results showed that the growth-promoting effect observed with B4 could not be reproduced by either of the closely related strains or by other additional members. Moreover, the presence of B4 specifically enhanced tetracycline degradation. These results suggest that the enhancing effect is unique to B4 rather than a general outcome of adding a third strain to the consortium.

On Lines 343-344, we write:

“...Furthermore, the potentiator strain B4 exhibits uniqueness: we tested species from the same genus or family, yet no such potentiating effect was observed (Fig. S11).”

On Fig. S11, we write:

“

Fig. S11 | Community member substitution test. Different bacterial consortia were inoculated into MM-peptone medium containing 10 ppm TC and OTC and incubated for 27 h: CP (the combination of strains *Comamonas* sp. C2 and *Providencia* sp. P1), CPB (the combination of strains *Comamonas* sp. C2, *Providencia* sp. P1, and *Brevundimonas* sp. B4), CPB-1 (the combination of strains *Comamonas* sp. C2, *Providencia* sp. P1, and *Brevundimonas vesicularis*, a strain of the same genus but different species from B4), and CPB-2 (the combination of strains *Comamonas* sp. C2, *Providencia* sp. P1, and *Caulobacter vibrioides*, a bacterium from the same family but different genus from B4). The optical density and degradation rate of TC were measured after 27 h of cultivation. a, Growth curves of different consortia. b, Degradation rates of TC different consortia. TC, tetracycline; OTC, oxytetracycline. The data are presented as mean values ($n = 4$ biological independent replicates).”

- In silico sensitivity test: Provide a brief GSMM sensitivity analysis (e.g., $\pm 20\%$ on exchange costs or carbon availability) to show robustness of model-driven recommendations.

Answer: We sincerely thank Reviewer 1 for this valuable suggestion. In response, we performed a sensitivity analysis on the carbon source composition to evaluate the robustness of our results with respect to the specific four-carbon mix (glucose, citrate, acetate, fumarate). Specifically, we used a baseline medium containing 20 mM glucose, 20 mM acetate, 20 mM citrate, and 20 mM fumarate, and inoculated it with the microbial consortium (strains A1, P1, C2, L3, S4, B4). Based on this baseline, we designed two types of perturbation experiments:

Single-factor perturbation: To assess the effect of fluctuations in individual carbon sources, we

adjusted the concentration of one carbon source by $\pm 10\%$ (i.e., increased or decreased by 10%) while keeping the others constant.

We calculated the relative biomass change as:

$$\text{Relative change rate} = (\text{Biomass_perturbed} - \text{Biomass_baseline}) / \text{Biomass_baseline}$$

and defined the robustness coefficient as $R = 1 - |\text{Relative change rate}|$ (where the absolute value accounts for both increases and decreases in biomass).

Multi-factor perturbation: To assess robustness under combined fluctuations, we simultaneously perturbed the concentrations of all four carbon sources by $\pm 10\%$ (consistent with the amplitude used in single-factor tests).

In both scenarios, the system was considered robust if the relative change rate remained $< 10\%$ and the robustness coefficient (R) was close to 1. Our results (summarized in Supplementary Table S8) showed that under all perturbation conditions, the relative biomass change rate ranged from -8.67% to 8.58% (all $< 10\%$), and R values were between 0.9133 and 0.9985 (close to 1).

In the model, we also conducted a robustness test: we set the carbon source content to fluctuate by $\pm 20\%$, and used SuperCC to simulate and predict the optimal biomass of the six-bacterium community model. Our results (summarized in Supplementary Table S9) showed that under all perturbation conditions, the relative biomass change rate ranged from -0.03% to 0.07% (all $< 1\%$), and R values were close to 1. These findings confirm that our results are robust to variations in the specific four-carbon mix used.

On Supplementary Table S8, we write:

“Supplementary Table S8 Experimental validation of sensitivity analysis for carbon source composition. Bacterial growth was monitored for 1800 min.

Condition (Carbon sources) mM				Mean Biomass	SD	Relative Change rate (%)	Robustness Coefficient (R)
Glucose	Acetate	Citrate	Fumarate				
20	20	20	20	0.7301	0.0160	–	–
22	20	20	20	0.7094	0.0243	-2.84%	0.9716
18	20	20	20	0.7326	0.0224	+0.34%	0.9966
20	22	20	20	0.7280	0.0315	-0.29%	0.9971
20	18	20	20	0.7928	0.0335	+8.58%	0.9142
20	20	22	20	0.6668	0.0523	-8.67%	0.9133
20	20	18	20	0.6942	0.0424	-4.92%	0.9508
20	20	20	22	0.7752	0.0441	+6.17%	0.9383
20	20	20	18	0.7078	0.0172	-3.06%	0.9694
22	22	18	18	0.7170	0.0232	-1.80%	0.982
18	18	22	22	0.7266	0.0246	-0.48%	0.9952
22	18	18	22	0.7030	0.0196	-3.72%	0.9628
18	22	22	18	0.7312	0.0181	+0.15%	0.9985

Note: Relative change rate = $(\text{Biomass_perturbed} - \text{Biomass_baseline}) / \text{Biomass_baseline} \times 100\%$

and defined the robustness coefficient as $R = 1 - |\text{Relative change rate}|$ (where the absolute value accounts for both increases and decreases in biomass). ”

On Supplementary Table S9, we write:

“Supplementary Table S9 Modeling validation of sensitivity analysis of DHP-Com consortium for carbon source composition.

Condition (Carbon sources)mmol/g DW				Optimal Biomass	Relative Change rate (%)	Robustness Coefficient (R)
Glucose	Citrate	Acetate	Fumarate			
25	25	25	25	421.812654823115	-	-
20	25	25	25	421.672057879753	-0.033331609%	0.999666684
30	25	25	25	421.953251766474	0.033331609%	0.999666684
25	20	25	25	421.812654823114	-2.42568E-13%	1
25	30	25	25	421.765789175328	-0.011110536%	0.999888895
25	25	20	25	421.765789175328	-0.011110536%	0.999888895
25	25	30	25	421.859520470903	0.011110536%	0.999888895
25	25	25	20	421.718923527538	-0.022221072%	0.999777789
25	25	25	30	421.906386118687	0.022221072%	0.999777789
20	20	30	30	421.812654823113	-4.7166E-13%	1
30	30	20	20	421.812654823071	-1.04304E-11%	1
20	30	20	30	421.718923527539	-0.022221072%	0.999777789
30	20	30	20	421.906386118687	0.022221072%	0.999777789
20	30	30	20	421.625192231966	-0.04442145%	0.999555579
30	20	20	30	422.000117414260	0.04442145%	0.999555579
20	20	20	20	421.531460936395	-0.066663217%	0.999333368
30	30	30	30	422.093848709832	0.066663217%	0.999333368

Note: Except for the change in the content of carbon source C, the contents of other inorganic substances remain unchanged. In particular, the content of NH_4^+ is always maintained at 100 mmol/g DW. ”

- Some bounding of the generalization claims is probably appropriate. If results are shown in one medium/regime, either (i) replicate a key result under one orthogonal condition or (ii) narrow the language to the tested context.

Answer: We thank the reviewer for their valuable suggestion. To better bound our generalization claims and align with the requirement of narrowing conclusions to the tested context, we have revised the manuscript. Specifically, we have uploaded the detailed composition and type information of all culture media used in the experiments to GitHub (<https://github.com/ruanzhepu/superCC.git>), with each medium clearly annotated to indicate the specific experiment it was employed for. This allows readers to fully contextualize our results within the exact experimental conditions tested.

METHODOLOGY

- Generally sound experimental and modeling choices.
- Units/quantification: Standardize growth/flux units and annotate axes; state detection limits explicitly. Replace “complete removal” with “< LOD (method, value)”.

Answer: We thank the reviewer for this valuable suggestion. We have standardized all growth and flux units throughout the manuscript and annotated the axes in all figures accordingly. The limits of detection (LOD) have been added to the pollutant analysis section in the Supplementary Materials.

On Supplementary Text, we write:

“Quantification of tetracycline and oxytetracycline by means of LC-MS

We conducted a quantitative analysis of tetracycline (TC), oxytetracycline (OTC), dodecyl trimethyl ammonium chloride (DTAC), dodecyl dimethyl benzyl ammonium chloride (DBAC), sulfamethoxazole (SMX), and sulfamethazine (SMT) using a liquid chromatography-mass spectrometry (LC-MS) system (Agilent 6470B). Chromatographic separation was achieved on a C18 reversed-phase column (150 mm × 4.6 mm, 5 μm particle size).

For TC and OTC, the injection volume was 2 μL. Mobile phase A consisted of 0.1% formic acid

in water, prepared using a Milli-Q Advantage ultrapure water system, while mobile phase B was HPLC-grade acetonitrile (purchased from Shanghai Anpu Co., Ltd.). The flow rate was maintained at 0.3 mL/min. The elution gradient was as follows: 0-5 min, 90% to 60% mobile phase A; 5-5.1 min, 60% to 90% mobile phase A; and 5.1-7.1 min, 90% mobile phase A. The column temperature was set at 30°C. Mass spectrometry was performed using an electrospray ionization (ESI) source with Agilent Jet Stream technology in positive ion mode. The scan type was set to multiple reaction monitoring (MRM) with a Delta EMV (+) of 300. The MRM parameters for TC were as follows: precursor ion 445.1 m/z, quantifier ion 410.1 m/z, qualifier ion 154 m/z, with a fragmentation voltage of 150V. Collision energy (CE) for ion 410.1 was set to 21V, and for 154 to 29V, with a cell acceleration voltage of 4V and positive polarity. For OTC, the precursor ion was 461.2 m/z, quantifier ion 426.1 m/z, and qualifier ion 443.1 m/z, with a fragmentation voltage of 125V and CE of 21V and 13V. The cell acceleration voltage was 4V, also in positive polarity. The retention times for TC and OTC were 2.6 min and 2.4 min, respectively. The limits of detection (LOD) were 0.12383 ppb for TC and 8.47458 ppb for OTC.

For DTAC and DBAC, the injection volume was 5 µL. Mobile phase A was 0.1% formic acid in water, and mobile phase B was acetonitrile. The flow rate was 0.3 mL/min. The elution gradient was: 0-1.5 min, 55% to 20% mobile phase A; 1.5-3 min, 20% to 0% mobile phase A; 3-6 min, 0% mobile phase A; and 6-10 min, 55% mobile phase A. The column temperature was 40°C. The MRM parameters for DBAC were as follows: precursor ion 304.3 m/z, quantifier ion 91.2 m/z, qualifier ion 212.2 m/z, with a fragmentation voltage of 125V. CE for ion 91.2 was set to 40V, and for 212.2 to 20V, with a cell acceleration voltage of 4V and positive polarity. For DTAC, the precursor ion was 228.3 m/z, quantifier ion 60.4 m/z, and qualifier ion 71.3 m/z, with a fragmentation voltage of 125V and CE of 24V and 28V. The cell acceleration voltage was 4V, also in positive polarity. The retention times for DBAC and DTAC were 2.1 min and 1.6 min, respectively. The LOD values were 0.00739 ppb for DBAC and 0.01044 ppb for DTAC.

For SMX and SMT, the injection volume was 2 µL. Mobile phase A was 0.1% formic acid in water, and mobile phase B was acetonitrile. The flow rate was 0.3 mL/min. The elution gradient was: 0-0.5 min, 90% mobile phase A; 0.5-1 min, 90% to 10% mobile phase A; 1-2.5 min, 10% mobile phase A; and 2.5-3 min, 10% to 90% mobile phase A. The column temperature was 35°C. The MRM parameters for SMT were as follows: precursor ion 279 m/z, quantifier ion 186 m/z, qualifier ion 92.1 m/z, with a fragmentation voltage of 109V. CE for ion 186 was set to 17V, and for 92.1 to 35V, with a cell acceleration voltage of 4V and positive polarity. For SMX, the precursor ion was 254 m/z, quantifier ion 92 m/z, and qualifier ion 156 m/z, with a fragmentation voltage of 109V and CE of 32V and 16V. The cell acceleration voltage was 4V, also in positive polarity. The LOD values were 0.02244 ppb for SMX and 0.00431 ppb for SMT.

For BO, the injection volume was 2 µL. Mobile phase A was 0.5% acetic acid in water, and mobile phase B was acetonitrile. The flow rate was 0.3 mL/min. The elution gradient was: 0-2 min, 20% mobile phase A. The column temperature was 30°C. The MRM parameters for BO were as follows: precursor ion 403 m/z, quantifier ion 300 m/z, qualifier ion 344 m/z, with a fragmentation voltage of 195V. CE for ion 300 was set to 32V, and for 344 to 28V, with a cell acceleration voltage of 4V and positive polarity. The LOD for BO was 0.0728 ppb.

Quantification of pollutants by means of HPLC

The analysis of bromoxynil octanoate (BO) and quinclorac (QC) was performed using a

high-performance liquid chromatography (HPLC) system (Agilent 1260). Detection of BO was facilitated through its intermediate metabolite, bromoxynil (Bxn).

For the chromatographic separation of Bxn, a C18 reversed-phase column (150 mm × 4.6 mm, 4 μm particle size) was employed. Bxn was detected at a wavelength of 250 nm. The column temperature was maintained at 40°C, and the injection volume was set to 20 μL. The mobile phase consisted of a mixture of acetonitrile/water/acetic acid (50/49.5/0.5, v/v/v), with a flow rate of 1.0 mL/min. The chromatographic separation of quinclorac was achieved using a C18 reversed-phase column (250 mm × 4.6 mm, 5 μm particle size). QC was detected at a wavelength of 240 nm. The column temperature was maintained at 30°C, and the injection volume was set to 10 μL. The mobile phase for quinclorac was a mixture of methanol/water/phosphoric acid (75/24.8/0.2, v/v/v), with a flow rate of 1.0 mL/min. The retention times for Bxn and quinclorac were 3.5 minutes and 3.9 minutes. The LOD were 180 ppb for Bxn and 6.6 ppb for QC.”

- **Reporting:** Ensure all figure panels referenced in text (e.g., 1a/1b/3c) are explicitly called out.

Answer: Thanks for the reviewer’s reminder regarding figure panel referencing. We confirm that all figure panels cited in the text (e.g., 1a/1b/3c) have been explicitly and clearly labeled throughout the manuscript.

On Lines 83 and 95, we write:

“...under compound pollution (Fig. 1a). Our work...”

“...and a combination of both (TCs) (Methods, Fig. 1a).”

On Line 98, we write:

“...compared to the single-pollutant groups (one-way ANOVA, $p = 0.00098$; Fig. 1b).”

On Line 143, we write:

“...*Stenotrophomonas* sp. S4 (Fig. 3a, 3c).”

ORGANIZATION AND CLARITY

- Bridge early analyses to the six focal isolates. Add a short paragraph/schematic explicitly linking Figs 1–2 to species selection (the focus of the rest of the paper).

Answer: Thanks for the reviewer’s comments. In line with the storyline of the results, we have supplemented Figure 1a with a more detailed analysis workflow, including the modeling of the Top 50 strains and the definition of the "potentiator". Corresponding explanatory notes for coherence have also been added in both the main text and the figure legend.

On Lines 694-696, we write:

“Meanwhile, systematic analysis was conducted on the metabolic models of the top 50 strains in terms of abundance in each treatment group to determine the growth and metabolic characteristics of the dominant species.”

On Lines 701-702, we write:

“Since metabolomic analysis was only performed on cocultures with up to three strains, the microbial ... models with a larger number of strains, aiming to predict ...”

On Fig. 1, we write:

“

Fig. 1 | Experimental design and microbial consortium responses to combined antibiotic pollution. *a*, The workflow involves the domestication of aquatic microbial communities from the same water sample using three acclimation approaches: exposure to tetracycline (TC) alone, oxytetracycline (OTC) alone, or a combination of both (TCs). Functional microbiomes were subsequently obtained and analyzed via amplicon sequencing. Meanwhile, systematic analysis was conducted on the metabolic models of the top 50 strains in terms of abundance in each treatment group to determine the growth and metabolic characteristics of the dominant species. Potential keystone species were thus identified through an integrated approach combining abundance shifts and strain isolation. To assess strain-specific metabolic responses under varying stress conditions, untargeted metabolomics was conducted on single-, co-, and tri-strain cocultures under three treatments: antibiotic-free control (CK), exposure to a single antibiotic (TC or OTC), and exposure to the combined TCs treatment. Since metabolomic analysis was only performed on co-cultures with up to three strains, the microbial community modeling framework SuperCC was applied to construct multi-strain metabolic models with a larger number of strains, aiming to predict strain-specific biomass and identify key interspecies metabolites. Model predictions were subsequently validated through experimental analyses. Based on the above conclusions, we proposed a synthetic microbiome construction strategy: the Degradator–Helper–Potentiator Consortium (DHP-Com) paradigm, and provided the definition of “potentiator”...”

• Figure hygiene: The paper felt a bit like a stream of consciousness. I'd start with a “Results map” table linking each scientific question to the figure/panel that answers it and drive the paper through more of a narrative style. It reads more like a huge dump of a tremendous amount of (good) methodology and analysis.

Answer: Thanks for the reviewer’s comments. To address the concern about the paper’s flow, we have added connecting sentences directly to each subsection of the Results section. These sentences serve to bridge different parts of the results, enhancing the overall logical coherence of the scientific narrative and making the story more cohesive.

On Lines 113-118, we write:

“The observed changes in community structure and function raised a follow-up question: which species within the consortium drive these traits? We focused on the Top-50 most abundant bacteria in each consortium, as these are likely key contributors to ecosystem function. Since the consortia shared the same origin, the Top-50 strains overlapped substantially, resulting in 70 unique strains after deduplication (Table S1). We constructed single-strain metabolic models to characterize their ecological strategies under complete medium (CM)...”

On Lines 140-141, we write:

“To translate these community-level patterns into practical applications (e.g., streamlined degrading consortia), we next isolated key strains from the three consortia. We obtained six strains...”

On Lines 166-167, we write:

“The synergistic growth and degradation of keystone species suggested they rely on metabolite-mediated interactions. To test this, we performed...”

On Lines 196-197, we write:

“To deepen our understanding of metabolite-mediated interactions, we used the...”

On Lines 225-226, we write:

“Finally, we tested whether keystone species and their metabolites could enhance resilience under real-world, multi-pollutant stress. We exposed the...”

BOTTOM LINE

Considerable revisions needed. Of particular note: (1) operational role definitions with a reported potentiator index of some kind (2) improved bridging/organization, and (3) the small ablation and in-silico sensitivity checks. If these are addressed thoroughly in text/analysis (and the vitamin mechanism is framed cautiously), I would be comfortable moving to acceptance.

Answer: Thanks for the reviewer’s valuable comments and constructive suggestions, which are of great significance for improving the quality and rigor of our manuscript. All conclusions in the revised manuscript are now supported by more comprehensive data and analysis. We sincerely hope you will find our revisions satisfactory and that the manuscript is suitable for publication in Nature Communications.

Reviewer's comments:

Reviewer #3 (Remarks to the Author):

The revision is improved and sound. The main contribution is methodological—using multistrain GSMM/FBA to guide role-based community design—with experimental results that broadly align with model predictions. The work is significant as an analysis/toolkit advance for engineering multi-member consortia under mixed-pollutant conditions. I support publication.

Answer: We thank the reviewer for supportive comments and for recommending our work for publication. We are pleased that the reviewer finds our methodological contribution significant and our results sound. We have carefully addressed all the specific points raised below to further strengthen the manuscript.

Reviewer #3 (Remarks on code availability):

Potentiator index: State the definition explicitly as the DH vs. DH+P contrast (or justify any difference), pre-specify one primary performance metric (growth AUC or % removal at fixed time), and report CIs.

Answer: Thank you for the thoughtful suggestions regarding the definition and reporting of the potentiator index. We have revised the manuscript to address all points raised:

Explicit definition of the Potentiator Contribution Index (PCI)

We now explicitly define PCI as the contrast between the Degradator+Helper (DH) condition and the Degradator+Helper+Potentiator (DH+P) condition:

$$PCI = \frac{\text{Performance}_{DH+P} - \text{Performance}_{DH}}{\text{Performance}_{DH}}$$

Here, Performance refers to the pollutant degradation rate or biomass of the consortium. This definition has been added to the main text (Lines 85–86) and incorporated into Table 1.

On Table 1, we write:

“**Table 1** Characteristics of Degradator, Helper, Potentiator, and Potentiator Contribution Index (PCI) in DHP-Com.

Table 1 Characteristics of Degradator, Helper, Potentiator, and Potentiator Contribution Index (PCI) in DHP-Com.

	Abundance	Significant difference in changes	Function	In this study
Degradator	Not necessarily high	Yes	Direct function, degrade pollutants	Strain C2
Helper	Not necessarily high	Yes	Indirect function, help degraders	Strains A1, P1, L3
Potentiator	High	No	Indirect function, help degraders and helpers	Strains B4, S4 (The abundance is usually among the top 30.)
Potentiator Contribution Index (PCI)	$PCI = \frac{\text{Performance (DHP)} - \text{Performance (DH)}}{\text{Performance (DH)}} \times 100\%$			

Performance (DHP) = Overall performance (e.g., pollutant degradation rate, community biomass) of the consortium containing Degradator (D), Helper (H), and Potentiator (P).

Performance (DH) = Performance of the consortium with only Degradator (D) and Helper (H). ”

On Lines 85-86, we write:

“...and also introducing a Potentiator Contribution Index (PCI) to quantitatively measure the potentiator’s effect across conditions (Table 1).”

We thank the reviewer for the suggestion. We note that PCI is intended to capture changes in both degradation rate and biomass, which may not always vary in a strictly linear or proportional manner. As a result, conventional confidence intervals may not be directly applicable. Accordingly, PCI is presented as a descriptive index to reflect the potentiator’s effect across conditions, rather than as a formal statistical estimate with confidence bounds.

Statistics: Surface the multiple-testing approach (e.g., FDR) in main text/captions and pair comparative statements with named tests and effect sizes/CIs.

Answer: We thank the reviewer for the suggestion. In the manuscript, we have aimed to clearly describe the statistical approach.

On lines 120-122, we write:

“biomass production correlated positively with growth time (Student's t-test, $p = 0.0107$, Fig. 2a), total metabolic reactions (Student's t-test, $p < 0.0001$, Fig. 2b), and exchange reactions (Student's t-test, $p < 0.0001$, Fig. 2c).”

On lines 184-185, we write:

“To identify key secreted metabolites, we applied a more stringent significance threshold (Student's t-test, $p < 0.01$, VIP value ≥ 2.00)”

On lines 243-244, we write:

“Keystone strains (TCs-enriched) demonstrated significantly enhanced growth when supplemented with predicted metabolites (one-way ANOVA, $p < 0.01$).”

On lines 769-774, we write:

“b, Quantitative analysis of significantly upregulated metabolites (compared to CK; Student's t-test, $p < 0.05$, VIP ≥ 1 in OPLS-DA), displaying metabolite counts (represented by color-coded circles) and mean fold-change ($\log_2FC \pm SEM$). The microbial consortium abbreviations (APCLSB) in the figure correspond to strains A1, P1, C2, L3, B4, and S4, respectively. Data are presented as mean \pm SD; $n = 4$ biological replicates. c, Functional characterization of metabolites under combined antibiotic stress (compared to CK; Student's t-test, $p < 0.01$, VIP ≥ 2 in OPLS-DA).”

Scope: Bound generalization claims to the tested medium/regime (or point clearly to any added orthogonal replication).

Answer: We thank the reviewer for the suggestion. To appropriately bound our generalization claims, we have clarified in the Supplementary Text (“Cultivation Media”) the specific media used for each experimental purpose.

In Supplementary Methods (1.1 Cultivation Media),we write:

“...The LB medium was used for strain activation and large-scale cultivation. The EM medium was

used for consortium acclimation, while the DM medium was employed to evaluate the degradation performance of the consortium. The MM medium was used for metabolomic experiments; MM-10% P was used for metabolite supplementation validation experiments; MM-P was used to assess the degradation performance of individual strains and to verify the loss-of-function of enhancer strains. The MM-G medium was used to determine the growth curves of the top 50 strains from the TCs subset.”

Mechanism wording: Keep vitamin/cofactor effects as “consistent with” unless a minimal auxotrophy/complementation or uptake-inhibition test is included.

Answer: We thank the reviewer for the suggestion. All references to vitamin or cofactor effects have been revised to “consistent with a role of” to avoid implying direct causation.

On lines 307-310, we write:

“Our observations are consistent with a role of vitamins in microbial growth and metabolism^{21,24,29}. In synthetic microbiomes, vitamin supplementation was associated with increased microbial biomass and enhanced pollutant removal. ”

Sensitivity: Add a sentence confirming whether rankings/index values remain stable under the reported parameter perturbations.

Answer: We thank the reviewer for this suggestion. We have added a clear sentence regarding the stability of our key findings.

On lines 467-468, we write:

“The relative change in biomass remained below 10% under all perturbations. These findings confirm that our results are robust to variations in the specific four-carbon mix used.”